# Unifying Predictions of Deterministic and Stochastic Physics in Mesh-reduced Space with Sequential Flow Generative Model

**Luning Sun**\*
Lawrence Livermore National Lab
Livermore, CA 94550
sun42@llnl.gov

**Xu Han** \*
Tufts University
Medford, MA 02155
xu.han@tufts.edu

**Han Gao**
Harvard University
Cambridge, MA 02138
hgao1@seas.harvard.edu

**Jian-Xun Wang**
University of Notre Dame
Notre Dame, IN 46556
jwang33@nd.edu

**Li-Ping Liu**
Tufts University
Medford, MA 02155
liping.liu@tufts.edu

## Abstract

Accurate prediction of dynamical systems in unstructured meshes has recently shown successes in scientific simulations. Many dynamical systems have a nonnegligible level of stochasticity introduced by various factors (e.g. chaoticity), so there is a need for a unified framework that captures both deterministic and stochastic components in the rollouts of these systems. Inspired by regeneration learning, we propose a new model that combines generative and sequential networks to model dynamical systems. Specifically, we use an autoencoder to learn compact representations of full-space physical variables in a low-dimensional space. We then integrate a transformer with a conditional normalizing flow model to model the temporal sequence of latent representations. We evaluate the new model in both deterministic and stochastic systems. The model outperforms several competitive baseline models and makes more accurate predictions of deterministic systems. Its own prediction error is also reflected in its uncertainty estimations. When predicting stochastic systems, the proposed model generates high-quality rollout samples. The mean and variance of these samples well match the statistics of samples computed from expensive numerical simulations.

## 1   Introduction

Accurate prediction of long-term dynamics of physical systems is of great interest in many science and engineering fields. The classical simulation of a system heavily relies on a spatial/temporal discretization of the space and the numerical solution to a finite-dimensional algebraic system derived from the governing equation [1]. However, due to the multi-scale, stochastic nature of the complex physics and the complexity of the geometry, the simulation of large-scale real-time applications is extremely expensive in computation [2]. In recent years, deep learning models have been applied to predict rollouts of large and complex physical systems thanks to their flexibility and scalability [3]. Moreover, they can handle uncertainties and non-linearity in physical problems [4–8] more effectively than traditional methods.

Previous work has studied two types of complex dynamical systems: deterministic ones and stochastic ones. A deterministic system is often considered under a perfectly controlled experiment with exactly

---

\*Equal contribution.

37th Conference on Neural Information Processing Systems (NeurIPS 2023).

known PDE terms and initial conditions (IC) or boundary conditions (BC) [9, 10, 3, 11–13]. This type of system can be modeled by autoregressive predicting models. A stochastic system such as quantum mechanics [14] and statistical physics [15] has a stochastic rollouts. When there are stochastic forcing terms or IC/BC terms in the governing equations, researchers have developed various models for predicting the stochastic state variables, such as turbulence velocity and stock prices [16–20]. Given that there is not a clear boundary between the two types of systems, it is highly desirable that a unified model can model either a deterministic or a stochastic system automatically. However, such models so far are limited to classical numerical models such as OpenFOAM [21]. There is an urgent need to develop a unified deep-learning model that can model both types of systems.

In this work, we propose a unified framework based on deep generative models to predict deterministic and stochastic systems. The new model is based on graph-structured state representations [11], which can handle irregular spatial areas commonly seen in dynamical systems. It uses an autoencoder to encode a state representation to a low-dimensional latent vector. Accurate encoding and decoding is critical for recovering the state of the dynamical system. This work makes several innovations to enhance the autoencoder's ability to preserve the information in system states. We provide a new approach to encoding the graph-structured representation with a vector with a fixed length. We also get some inspiration from regeneration learning [22] and train our autoencoder with self-supervised learning.

To describe stochasticity in the system, we use a sequential probabilistic model for the latent representations. We integrate a transformer and a normalizing flow to construct a step-wise predictive neural network in the latent space. When there is stochasticity, the model will learn the conditional distribution of the next latent state; when there is no stochasticity, it can place probabilities to correct deterministic predictions and still can minimize the predictive error. By including the ability to simulate both deterministic and stochastic systems in a uniform framework, it reduces the effort of developing separate models for different problems.

We evaluate our proposed framework in an extensive empirical study. The results indicate the proposed model outperforms the SOTA baselines on deterministic datasets regarding accuracy. More importantly, for the first time, we introduce several alternative evaluation metrics other than normalized RMSE for stocahstic fluid dynamics, which help improve comparisons between different methods in this domain. The proposed framework can produce high-quality samples for stochastic systems in the mesh space.

## 2   Background

### 2.1   Problem definition

Let's consider a general partial differential equation (PDE) defined on the $d$-dimensional space and one-dimensional time domain,

$$\frac{\partial \boldsymbol{u}}{\partial t} = j(\boldsymbol{u}, \boldsymbol{\mu}, \boldsymbol{\iota}) \quad \text{in } \Omega \times [0, T_{\text{end}}] \tag{1}$$

where $\Omega \subset \mathbb{R}^d$ is the spatial domain, $T_{\text{end}}$ is the endpoint of time, $\boldsymbol{\iota} : \partial\Omega \times [0, T]$ is the random parameter over time for stochastic systems (e.g., boundary conditions), and $\boldsymbol{u} : \Omega \times [0, T]$ is the primary solution variable (e.g., velocity and pressure of fluid flow). Here $\boldsymbol{\mu} : \mathcal{D}$ is the global physical system parameter and is time-invariant (e.g., Re number) for a given rollout. $j : \Omega \times [0, T] \times \mathcal{D} \to \Omega \times [0, T]$ is an aggregation of the spatial terms of conservation laws (e.g., source and flux). To numerically solve the conservation law in Equation 1, let $\mathcal{C}_h$ be a mesh of $\Omega$, that is, $\mathcal{C}_h = \{C_i \in \Omega : i = 1, \ldots, N\}$ is a collection of non-overlapping cells that cover $\Omega$. We further use $\boldsymbol{u}_{i,t}$ to denote the evaluation of $\boldsymbol{u}$ at the cell center of $C_i$ at time step $t$.

Using the mesh described above, we apply a finite volume discretization to yield the parametrized, nonlinear dynamical system. Here, the dynamical system can be computationally intensive because a fine-level mesh will lead to discretization with a large degree of freedom. For stability issues, the numerical time step also needs to be very small. Therefore, numerical simulations with traditional methods require extremely expensive computation and storage.

In this paper, we are interested in two different scenarios: *deterministic* dynamics with invariant parameter $\boldsymbol{\iota}$ over time (e.g., fixed boundary condition ), and *stochastic* dynamics with random

parameter $\boldsymbol{\iota}$ over time (e.g., perturbations in boundary conditions of turbulent flow). Both cases will also include the global physical parameters $\boldsymbol{\mu}$. The scope of our paper is to build a unified data-driven surrogate model for generating/predicting parametric PDE solutions.

## 2.2 Normalizing flow models

A normalizing flow model constructs a flexible probabilistic distribution by applying a learnable bijective mapping to a simple random variable (e.g. Gaussian distributed). Suppose the mapping is $\boldsymbol{z} = f(\boldsymbol{x})$ with $\boldsymbol{x} \in \mathbb{R}^d$ being the simple input variable, then we have the probability $p(\boldsymbol{z})$ of $\boldsymbol{z}$ as follows:

$$p(\boldsymbol{z}) = p(\boldsymbol{x}) \left| \det \left( \frac{\partial(f(\boldsymbol{x}))}{\partial(\boldsymbol{x})} \right) \right|^{-1}. \tag{2}$$

Here $\frac{\partial(f(\boldsymbol{x}))}{\partial(\boldsymbol{x})}$ is the Jacobian of $f$ at $\boldsymbol{x}$. The function $f$ is usually a neural network with a layered structure, with each layer being a bijective mapping. Then the determinant of $\frac{\partial(f(\boldsymbol{x}))}{\partial(\boldsymbol{x})}$ is the product of determinants of these layers' Jacobian matrices. With a special design of layer structures, their determinants can be efficiently computed. For example, a RealNVP model [23] is constructed by stacking several *coupling* layers, each of which has a lower triangular Jacobian matrix. With $\boldsymbol{h} \in \mathbb{R}^d$ as the input, a coupling layer $f_\ell$ runs the following calculation:

$$f_\ell(\boldsymbol{h}) = \boldsymbol{concat}(\boldsymbol{h}[1:d'], \boldsymbol{h}[(d'+1):d] \odot \exp(s(\boldsymbol{h}[1:d'])) + t(\boldsymbol{h}[1:d'])) \tag{3}$$

Here $\boldsymbol{concat}$ concatenates its arguments as one vector, and $d'$ is usually about one-half of $d$. In this calculation, the first half of the vector is directly copied to the output. The second half goes through an entry-wise linear transformation: the operation $\odot$ is the Hadamard product, the neural network $s$ provides scaling coefficients, and the neural network $t$ provides biases. With $K$ such coupling layers, we have a transformation that defines a flexible distribution $p(\boldsymbol{z})$.

$$p(\boldsymbol{z}) = p(\boldsymbol{x}) \cdot \prod_{\ell=1}^{K} |\det(\partial f_\ell(\boldsymbol{h}^{\ell-1})/\partial \boldsymbol{h}^{\ell-1})|^{-1} \tag{4}$$

Here $\boldsymbol{h}^0 = \boldsymbol{x}$, and $\boldsymbol{h}^\ell = f_\ell(\boldsymbol{h}_{\ell-1})$. At the same time, $p(\boldsymbol{z})$ also has an efficient sampling procedure given by $f$.

## 3 Methodology

Our new predicting model has three components: an encoder that compresses a graph representation of the state into a fixed-length vector, a sequential model that predicts next-step representations in the latent space, and a decoder that decodes spatial states from graph representations. The encoder is an improved version of the GMR-GMUS encoder [11]. The sequential model is a conditional flow model. The encoder and decoder are trained via self-supervised learning as in regeneration learning.

### 3.1 Graph representation learning

Following GMR-GMUS, we use a graph $\mathcal{G} = (\mathcal{V}, \mathcal{E})$ to encode a snapshot of a deterministic or stochastic dynamical system at time step $t$. Here each node $i \in \mathcal{V}$ corresponds to the mesh cell $C_i$, and each edge $(i, j) \in E$ represents a neighboring relationship between two cells. At each step $t$, the solution $\boldsymbol{u}_{i,t}$ at cell $i$ becomes the feature vector of the node $i \in \mathcal{V}$ at time $t$. Because the mesh is pre-defined and fixed, the graph representation $(G, (\boldsymbol{u}_{i,t}, i \in V))$ preserves the full information of the mesh $\mathcal{C}_h$ at time $t$. Here we use $\boldsymbol{Y}_t = (\boldsymbol{u}_{i,t} : i \in V)$ to denote a snapshot at time $t$.

The key task for the encoder is to encode the graph into a low-dimensional vector $\boldsymbol{z}_t$. For this task, we use an improved version of the encoder of GMR-GMUS. The GMR-GMUS encoder runs a graph neural network over the graph representation to learn node representations. Then it takes node vectors of a selection of "pivotal" nodes and concatenates them to get $\boldsymbol{z}_t$. In our new encoder, we select a set of locations in the spatial space instead of graph nodes to aggregate spatial information: each selected location aggregates vectors of nearby nodes to get its representation. It decouples the graph representation and the aggregation operation so that a select location can encode nodes within an arbitrary distance. To improve the training stability, we also improve the graph neural network's

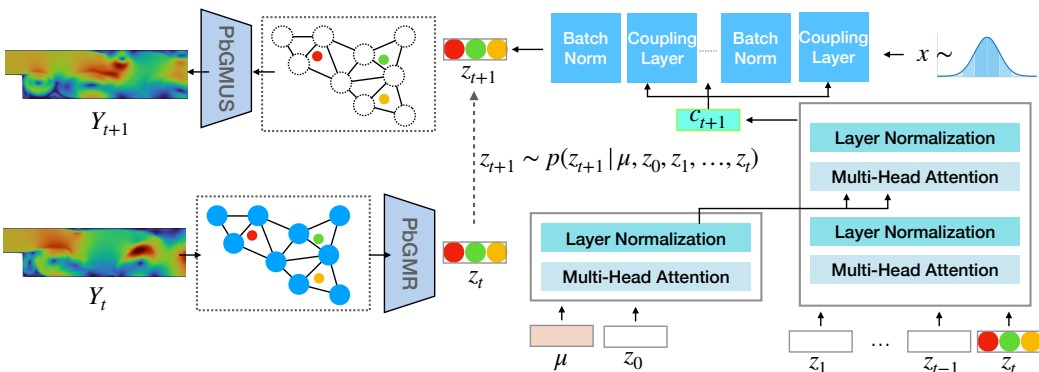

**Figure 1:** The diagram of the proposed model, which first compresses the whole graph into the latent representation $z_t$ by PbGMR using selected positions. During the generation process, a transformer encodes physical parameters and previous latent representations into a condition vector $c_{t+1}$, from which a normalizing flow model describes the conditional probability of $z_{t+1}$. Finally, the decoder PbGMUS decodes $z_{t+1}$ through to obtain the next-step prediction $Y_{t+1}$.

architecture by adding residual connections between its message-passing layers. We call this new encoder Position-based Graph Mesh Reducer (PbGMR). These two modifications clearly improve the encoder's ability to preserve the state information, which will be empirically shown in the experiment section.

The architecture of PbGMR is specified as follows. For notational convenience, we omit the time from the subscript. For each graph node $i \in V$, we first extract node and edge features as follows.

$$\boldsymbol{v}_i^0 = \mathrm{mlp}_v(\boldsymbol{u}_i), \quad \boldsymbol{e}_{ij}^0 = \mathrm{mlp}_e(\mathrm{pos}(i) - \mathrm{pos}(j)). \tag{5}$$

Here $\mathrm{pos}(i)$ is the spatial location of the cell center of $C_i$. After that, we apply $L$ message-passing layers.

$$\boldsymbol{e}_{ij}^\ell = \boldsymbol{e}_{ij}^{\ell-1} + \mathrm{layernorm}(\mathrm{mlp}_\ell^e\left(\boldsymbol{e}_{ij}^{\ell-1}, \boldsymbol{v}_i^{\ell-1}, \boldsymbol{v}_j^{\ell-1}\right)) \tag{6}$$

$$\boldsymbol{v}_i^\ell = \boldsymbol{v}_i^{\ell-1} + \mathrm{layernorm}\left(\mathrm{mlp}_\ell^v\left(\boldsymbol{v}_i^{\ell-1}, \sum_{j\in\mathcal{N}_i}\boldsymbol{e}_{ij}^{\ell-1}\right)\right), \quad \ell = 1,\dots,L. \tag{7}$$

Here $\mathcal{N}_i$ denotes all neighbors of node $i$, $\mathrm{mlp}_\ell^e$ and $\mathrm{mlp}_\ell^v$ are two separate multi-layer perceptrons, and $\mathrm{layernorm}$ is the layer-normalization operation. When a perceptron accepts multiple arguments, its arguments are first concatenated into a single vector. Each layer here is similar to a GraphNet block [24] but with slight differences. The prominent ones are residual connections and layer normalization in the update of node and edge representations: they help to stabilize the training procedure [25–27].

After we have learned node representations, we need to aggregate them into a single vector to represent the entire state. We randomly select a small set $\mathcal{S}$ of centers in the spatial area. For each center $c \in \mathcal{S}$, we select $k$ nearest mesh cells $\mathrm{knn}(c)$ based on spatial distance and then compute the position representation $\boldsymbol{h}_c$ by interpolation [28]:

$$\boldsymbol{h}_c = \sum_{j\in\mathrm{knn}(c)} \frac{w_{cj}\boldsymbol{v}_j^L}{\sum_{j\in\mathrm{knn}(c)} w_{cj}}, \quad w_{cj} = \frac{1}{d(c,j)^2}, \quad , c \in \mathcal{S} \tag{8}$$

Here $d(c,j)$ is the spatial distance between cell $C_j$ and the center $c$. Given that the calculation is scaled by the sum of $w_{cj}$-s, the unit of the spatial distance does not change the calculation here. The number $k$ of neighbors is a hyper-parameter, and we fix it to 10. Finally, we concatenate all representations of centers into a single vector $\boldsymbol{z} = \mathrm{concat}(\boldsymbol{h}_c : c \in \mathcal{S})$ as the latent for the entire graph. Note that the centers in $\mathcal{S}$ are fixed for a problem.

As a comparison, the encoder in GMR-GMUS uses graph nodes as centers and only considers connected neighbors of a center. Then the number of neighbors in the interpolation operation is

limited by the graph structure. As a result, each center vector from GMR-GMUS can only represent information in a short range. Our new encoder overcomes this issue by decoupling the neighbors in interpolation and the neighbors in the graph representation and then gives the interpolation operation more freedom to represent a state of the system.

## 3.2 Decoding and self-supervised training

We devise a decoder PbMUS to recover node features on the graph from latent representation $z$: $\hat{Y} = \text{PbGMUS}(z)$. Here we also consider the computation at a single time step and omit time indices. We first split $z$ and get vectors at interpolation centers: $(h_c : c \in \mathcal{S}) = z$ and then compute the initial node representation $r_i^0$ by spatial interpolation from centers:

$$r_i^0 = \sum_{j \in \text{knn}'(i)} \frac{w_{ic} h_c}{\sum_{c \in \text{knn}'(i)} w_{ic}}, \quad w_{ic} = \frac{1}{d(i,c)^2}, \quad , i \in \mathcal{V}, c \in \mathcal{S} \tag{9}$$

Here $\text{knn}'(i)$ are $k$ centers that are nearest to the cell center $i$. Then we apply $L$ message-passing layers to compute $\hat{Y} = \text{gnn}(G, (r_i^0 : i \in V))$, with gnn representing the $L$ network layers. These layers have the same architecture as PbGMR but use different learnable parameters.

**Self-supervised training.** Without considering the sequential property of the data, we first train the encoder and decoder on single steps with self-supervised training. This training method shares the same spirit of regeneration learning and improves encoder and decoder's abilities to capture spatial patterns in the data. In particular, we use the reconstruction error as the minimization objective to train the encoder-decoder pair.

$$\min \sum_{t=0}^{T} ||Y_t - \text{PbGMUS}(\text{PbGMR}(Y_t))||_2^2 \tag{10}$$

## 3.3 Attention-based temporal conditioned generative model

Now we consider the sequence of latent representations from the encoder and devise a sequential generative model to model the sequence of latent representations. The low-dimensional latent space reduces the modeling difficulty, and the vector form of latent representations $z_t$ avoids the graph structure and allows more model choices. We first decompose the sequence as follows.

$$P(z_{1:T}|\mu, z_0) = p(z_1|\mu, z_0) \prod_{t=2}^{T} p(z_t|\mu, z_0, z_{1:t-1}). \tag{11}$$

The key is to devise a model for the conditional $p(z_t|\mu, z_0, z_{1:t-1})$. We take two steps to construct this conditional: we first represent the condition $(\mu, z_0, z_{1:t-1})$ with a single vector $c_t$ and then adapt RealNVP [23], a normalizing flow model, to describe the conditional $p(z_t|c_t)$.

**Calculate the conditional vector with a transformer.** Because physical parameters $\mu$ control the entire system, and the initial condition $z_0$ contains information about the initial condition, we consider them special and use them in the prediction of $z_t$ for each $t$. We structure the problem as a "translation" problem and use the transformer [29] to run the calculation. In our case, the input "sentence" is $(\mu, z_0)$, the first few "tokens" in the target sentence are $(z_1, \ldots, z_{t-1})$, and the "next token" to be predicted is $c_t$.

$$c_t = \text{transformer}((\mu, z_0), (z_1, \ldots, z_{t-1})) \tag{12}$$

While the computation is exactly the same as one step in a translation task, the rationale is very different. First, the translation task directly gets the $z_t$ from the transformer, but our model needs to send $c_t$ to a conditional flow model to get the prediction $z_t$. Second, the translation task has an informative input sequence, but our model uses a less informative input and depends on the transformer to get a reasonable output sequence.

**Predict $z_t$ with a conditional flow model.** Once we have a vector $c_t$ representation of the condition, we can construct a conditional flow model from RealMVP. Specifically, we append the condition vector $c_t$ to the input of the scaling function $s$ and the bias function $t$ in equation 3 in each layer.

$$f_\ell(\boldsymbol{h}) = [\boldsymbol{h}[1:d'], \boldsymbol{h}[d'+1:d] \odot \exp(s(\boldsymbol{h}[1:d'], \boldsymbol{c}_t)) + t(\boldsymbol{h}[1:d'], \boldsymbol{c}_t)] \qquad (13)$$

From these layers, we have the flow model $p(z_t|c_t)$. By chaining up the transformer and the conditional flow model, we have our sequential model for conditional probability $p(z_t|\boldsymbol{\mu}, z_0, z_{1:t-1})$.

During training, we train the sequential model on $(z_0, \ldots, z_T)$ that are computed from our PbGMR encoder. During inference, we can efficiently sample $z_t$ from the sequential model. The whole training and inference processes can be found in Appendix A.2 and Appendix A.3.

## 4 Related Work

**Deep learning for physics.** For a deterministic system, one-step learning simply takes the current state as the input and outputs the next-step prediction [3, 30–32]. To further improve the accuracy of long-term forecasts, dimension reduction is applied together with sequence nets, aiming to solve long dynamical systems on a regular domain [33–35]. Such works often adopt CNNs as encoders so that can not be applied to irregular mesh space. To further work on mesh data directly with dimension reduction, the GNN encoder/decoder was initially introduced by [11]. Various GNN architectures were also proposed to facilitate the learning of physics. [13, 15, 36–39]. For physical stochastic systems, CNN is mainly used for probabilistic prediction [18, 40]. However, it lacks researches solving stochastic system in the graph space.

**Regeneration learning and generative modeling.** Regeneration learning is a new learning paradigm for data generation, which first generates a latent representation for the data. Then the generation process happens on the latent space. There are many recently popular generative models for images[41–43], videos[44], speeches[45], and music [46] are built on this paradigm. We notice that it should be further investigated for graph generation tasks.

## 5 Experiments

### 5.1 Deterministic dynamics

We benchmark our proposed model with three datasets from three flows: flows over the cylinder, high-speed flows over the moving edge, and vascular flows [11]. Note that for test cases, we feed the model physical parameters $\boldsymbol{\mu}$ and the snapshot at the beginning time to generate the whole trajectory. Detailed description can be found in Appendix A.5.

We compare our new method against five SOTA models for fluid dynamics, including two variants of MeshGraphNet[3] and three variants of the GMR-GMUS[11]. We use relative mean square error (RMSE) as the evaluation metric: $\mathrm{RMSE} = \frac{\sum(\hat{u}_i^{\mathrm{prediction}} - \hat{u}_i^{\mathrm{truth}})^2}{\sum(\hat{u}_i^{\mathrm{prediction}})^2}$. More details can be found in the Appendix A.2. For cylinder and vascular flows, we calculate the RMSE with respect to velocity variables $u$, $v$, and the pressure variable $p$. For high-speed flows over the moving edge, we also calculated the RMSE for the temperature $T$.

**The results on prediction errors.** Table 2 shows the comparison of different models. Our model outperforms the baseline models in all the scenarios. For the cylinder flow case, our model has improvements of 22%, 17%, and 47% for $u$, $v$, and $p$. For the sonic flow case, our model has improvements of 61%, 70% for the $u$ and $v$ variables, and improvements of 82% and 50% respectively for the $p$ and $T$ variables. For the vascular flow case, our model achieves improvements of 52%, 45%, and 95% for $u$, $v$, and $p$. The ablation

**Table 1:** The average relative reconstruction error of three systems, with the unit of $1 \times 10^{-3}$

| Dataset | GMR-GMUS | PbGMR-GMUS |
|---------|----------|------------|
| Cylinder flow | 14.3 | **1.9** |
| Sonic flow | 1.11 | **0.24** |
| Vascular flow | 10 | **2.8** |

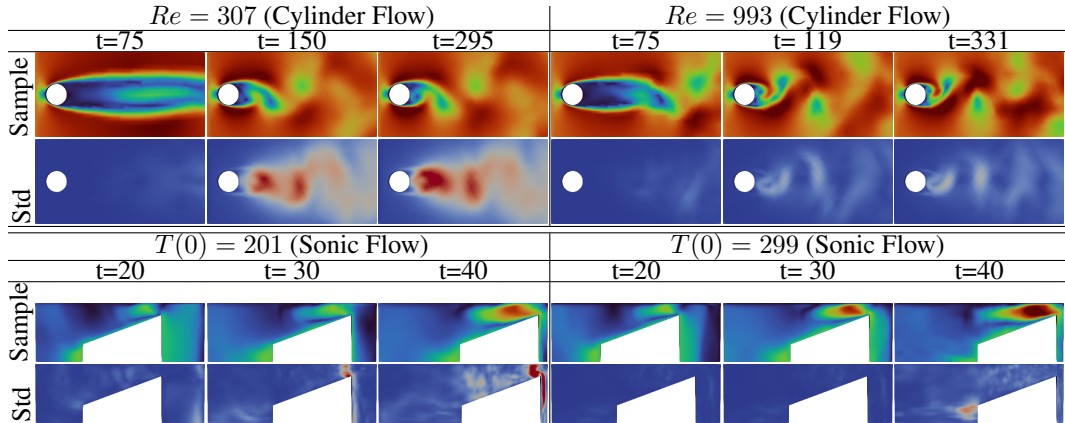

**Figure 2:** For each case, the first row is one of our predicted samples for velocity. The second row is the predicted standard deviation for velocity. Our model-predicted sample accurately reflects the physical systems. Meanwhile, the predicted standard deviation also has a physical spatial-temporal pattern.

study in the Appendix section (Table 9) indicates that both the new encoder and the conditional normalizing flow are essential to improve prediction accuracy.

The error in the time axis is shown in Figure 3. The performance of our model is shown with the solid black line. The model has small initial errors in all three tasks. Then its error accumulates over time, but the accumulation is much slower than that of competing models, so our model consistently has the lowest error in the final time step.

To better understand our model, we have also done an extensive ablation study reported in the appendix. Here we only show a comparison of our model with GMR-GMUS in the encoding-decoding tasks. The results in Table. 1 show that the new model has clearly better encoding and decoding capabilities. In one encoding and decoding case, the reconstruction error of our model is less than one fifth of that of the GMR-GMUS model in the appendix. The ablation study also shows that both residual connections and encoding centers enhance the encoding-decoding accuracy (please see Table 8 in the Appendix section).

**Contour analysis.**  As a probabilistic model, we plotted the contour of the predicted sample and the standard deviation (std) for all three datasets. Figure 2 visualizes the velocity predictions from two datasets. In each subplot, the top row is one prediction (e.g. one sample from our model), and the bottom row visualizes the standard deviation of predictions. To clearly compare different predictions, we use the same color bar within the same dataset. With visual inspection, the sample predictions are reasonable and stable for all three datasets.

The standard deviations also show reasonable spatial-temporal patterns consistent with our expectations. For cylinder flow with $Re = 307$, the standard deviation is larger around the vortex-shedding region, indicating the model is less certain about the fast-changing dynamics. Moreover, the standard deviation tends to grow with time, reflecting the error accumulation behavior for long-time rollouts. Moreover, the standard deviation is smaller when $Re$ is higher, indicating that the model is more certain about the prediction. The same trends can also be seen in other datasets. Therefore, our proposed model can accurately predict the dynamics for deterministic systems while providing a spatial-temporal uncertainty estimate, potentially improving the interpretability of deep learning systems. More contours can be found in Appendix A.6

### 5.2 Stochastic dynamics

**Dataset.**  We apply the proposed method to solve a stochastic dynamical system governed by the unsteady-state incompressible Navier–Stokes equations. We also compare our method to the unsteady Reynolds-averaged Navier–Stokes equations (URANS), the most applied method in the industry due to the balance of accuracy and computational efficiency. Let $\Omega$ be the channel with a backward-facing

**Table 2:** The average relative rollout error of three systems, with the unit of $1 \times 10^{-3}$. We compare with five different baselines. Two variances of MeshGraphNet with or without noise injection (NI) and three variants of GMR-GMUS (LSTM, GRU, or Transformer).

| Dataset-rollout step | | Cylinder flow-400 | | | Sonic flow-40 | | | | Vascular flow-250 | | |
|---|---|---|---|---|---|---|---|---|---|---|---|
| Variable | | $u$ | $v$ | $p$ | $u$ | $v$ | $p$ | $T$ | $u$ | $v$ | $p$ |
| MeshGraphNet [3] | NI | 25 | 778 | 136 | 1.71 | 3.67 | 0.4 | 0.027 | 57 | 133 | 55 |
| | without NI | 98 | 2036 | 673 | 4.12 | 6.13 | 0.24 | 0.020 | 3117 | 1771 | 601 |
| GMR-GMUS [11] | GRU | 114 | 1491 | 1340 | 1.34 | 4.59 | 0.59 | 0.37 | 8.2 | 11.2 | 23.6 |
| | LSTM | 124 | 1537 | 1574 | 1.57 | 5.8 | 0.69 | 0.45 | 8.4 | 11.1 | 23.3 |
| | decoding-only | 4.9 | 89 | 38 | 0.95 | 2.8 | 0.43 | 0.39 | 7.3 | 10 | 22 |
| PbGMR-GMUS | encoding-decoding | **3.8** | **74** | **20** | **0.37** | **0.85** | **0.079** | **0.01** | **3.49** | **5.47** | **1.05** |

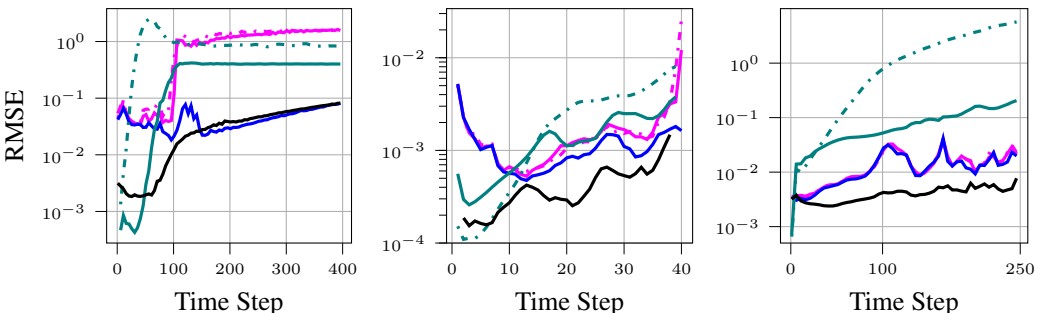

**Figure 3:** Averaged error over all state variables on cylinder flow (left), sonic flow (middle) and vascular flow (right), for the models MeshGraphNets(MGN)(· - · -), MGN-NI(——), GMR-GMUS-GRU (——), GMR-GMUS-LSTM (· - · -), GMR-GMUS-Transformer (——), Ours model(——).

step shown in Figure 5. The goal of the problem is to generate the temporal unsteady velocity field close to the ones simulated by large eddy simulation (LES) with the inflow velocity along the inlet boundary subject to uncertainty. The inflow at every time step is uncertain and sampled from a 60-dimensional stochastic space subject to a uniform distribution as $\{u_1^{\text{inlet}}, ..., u_{60}^{\text{inlet}} | u_i^{\text{inlet}} = 10 + u', u' \sim \mathcal{U}(0, 1)\}$.

**Analysis of fluid motion.** To visually study the generated results, we plot several time steps of the stream-wise velocity ($u$) from a collection of model samples in Figure 4. The proposed method clearly generates diverse turbulent flow samples. Specifically, the multiscale structure of the vortex can be seen in the contour plots. To quantitatively examine the model performance on turbulent statistics, we plot and compare the mean and variance of the velocity profiles from the URANS, the proposed model, and LES results in Figure 5. The mean velocity profile predicted by the URANS model (gray lines) has significant discrepancies compared to the LES result (cyan lines), particularly in the right half of the domain where the backward-facing step's geometry is less important and small-scale turbulent features are dominant. As for the velocity variance, the URANS model underestimates the fluctuation in the whole domain. On the contrary, both the LES velocity mean and variance can be captured well by our generated samples (purple lines).

**Comprehensive evaluations.** To evaluate the temporal energy transport, we plot the power spectral density using Welch's method [47] in Table 3 (*left*). The high wave numbers reflect the rapid changes of fluid motion (e.g., instantaneous fluctuations), and the low wave numbers reflect the slow changes, such as the re-circulation introduced by the backward-facing step. By leveraging the sequence models,

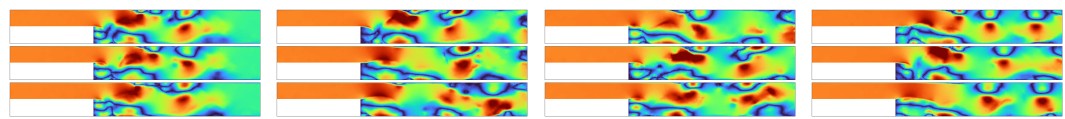

**Figure 4:** Velocity contour at time step $40, 80, 120, 160$ (*left to right*) from different samples.

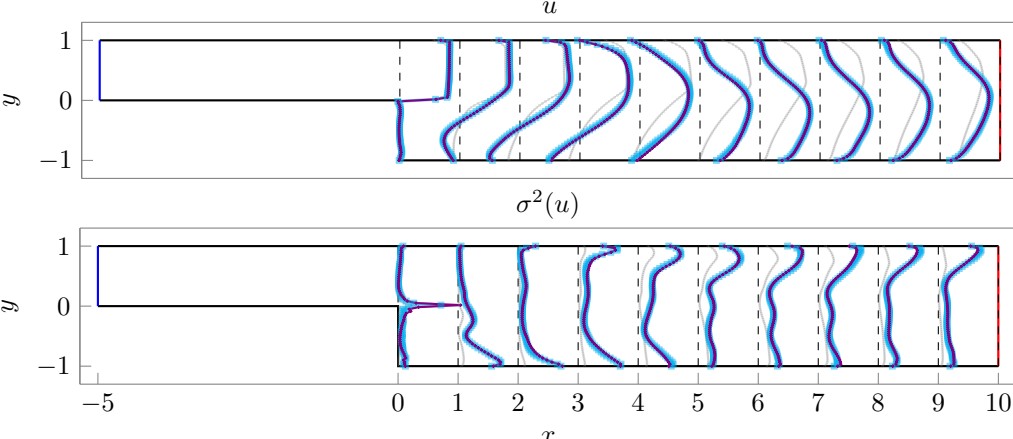

**Figure 5:** Evaluation lines: Inlet (——), Outlet (——), Wall (——), $x$-evaluation lines (- - -). Quantities from LES (——), URANS (——), and our model (——)

the multiscale problem in the time domain can be automatically learned. However, URANS directly calculate an ensemble average over infinite experiments, and it can not capture stochastic behavior. Therefore, the energy ($E(k)$) of high-wave-number ($k$) signals is underpredicted by URANS while the proposed model has a consistent energy spectrum pattern with the LES simulations. Moreover, the turbulent kinetic energy (TKE) is computed to evaluate the quality of generated samples. Physically, the TKE is characterized by measured root-mean-square velocity fluctuations. By applying the decoder, the point-wise instantaneous flow can be recovered from the latent vector to the physical solution reasonably well. The TKE metric can be defined as a scalar:

$$\text{TKE} := \frac{\int_\Omega \int_{[0,T_{\text{end}}]}[(u - \bar{u})^2 + (v - \bar{v})^2]d\Omega dt}{\int_\Omega \int_{[0,T_{\text{end}}]}[(u^{\text{les}} - \bar{u}^{\text{les}})^2 + (v^{\text{les}} - \bar{v}^{\text{les}})^2]d\Omega dt}. \tag{14}$$

It indicates the preservation of the TKE benchmarked by the LES simulations. Our model can generate a flow field preserving 99% of the energy, whereas URANS can only do less than 20%. As for the temporal mean field, although URANS aims to solve it directly, it is inevitable to introduce the bias from the LES result due to the ergodicity assumption of RANS. Instead, we directly formulate the probabilistic problem to learn the distribution of the spatiotemporal field. We calculate the error of the temporal mean field and find that although URANS leverages the conservation laws, it still obtains a much larger error than our generated samples. Finally, we further propose to measure the quality of predicted rollouts using evaluation metrics of video generation in computer vision. In particular, we consider two metrics: the continuous ranked probability score (CRPS) [48, 49] and Frechet Video Distance (FVD) [50]. We use CRPS to assess the respective accuracy of our probabilistic forecasting and LES models. Since our model considers the distribution of the forecasts as a whole, it outperforms URANS, which only focuses on the mean of the distribution. Our model is also better evaluated in the FVD metric. This indicates the learned distribution by our model is close to the samples from LES.

We have also experimented with two deep learning methods, MeshGraphNet and GMR-GMUS, which are designed for deterministic systems. Their predictions are both far from LES simulations, not to mention that they can only make deterministic predictions. More results can be found in Appendix A.13.

## 6 Conclusion

We propose a new learning model to predict/generate deterministic and stochastic fluid dynamics over unstructured mesh in a uniform way. With an integration of a novel graph auto-encoder, a transformer, and a normalizing flow model, the new model decomposes temporal and spatial correlations in a dynamical system. It outperforms competitive baseline models for deterministic systems while providing a reasonable spatial-temporal pattern of forward uncertainty estimations. The samples

**Table 3:** Energy cascade spectrum at $(2.58, 0.21)$(*left*), and criteria of generation quality(*right*).

| URANS (—●—), LES (—■—), Ours(—●—) | Quantity of interest | URANS | Ours |
|---|---|---|---|
|  | CRPS ($C_u$,↓) | 3.24 | **1.28** |
| | CRPS ($C_v$,↓) | 2.0 | **1.08** |
| | FVD ($d_u$,↓) | 228112 | **1262** |
| | FVD ($d_v$,↓) | 137860 | **397** |
| | mean flow error ($e_u$,↓) | 0.31 | **0.0176** |
| | mean flow error ($e_v$,↓) | 0.94 | **0.176** |
| | turbulent energy (TKE,↑) | 0.192 | **0.99** |

from the model trained on stochastic systems capture the rich physical patterns of expensive LES simulations. The current model still has one limitation: it can not accurately model stochastic variables close to boundary areas, which is shown in Appendix A.12. In future work, we will design new learning architectures to overcome this issue.

## Acknowledgement

We thank all reviewers for their insightful feedback. Liu was supported by NSF CAREER 2239869. Wang's group would like to acknowledge the funds from Office of Naval Research under award numbers N00014-23-1-2071 and National Science Foundation under award numbers OAC-2047127.

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

# A  Appendix

## A.1  Proof

The Jacobian of the transformation in Equation (4) can be written as:

$$\frac{\partial f^i(x)}{\partial f^{i-1}(x)} = \begin{bmatrix} \mathbb{I}^{d'} & 0 \\ \frac{\partial f^i(x)^{d'+1:d}}{\partial f^{i-1}(x)^{1:d'}} & \mathrm{diag}\big(\exp\big[s\big(f^{i-1}\mathrm{concat}((x)^{1:d'},c)\big)\big]\big) \end{bmatrix} \tag{15}$$

fwhere $\mathrm{diag}\big(\exp[s(f^{i-1}\mathrm{concat}((x)^{1:d'},c))]$ is the diagonal matrix whose diagonal elements correspond to the vector $\exp\big[s\big(f^{i-1}\mathrm{concat}((x)^{1:d'},c)\big)\big]$. Since the concatenation operation still keeps the matrix triangular, the determinant can be computed as:

$$\det(\partial f^i(\boldsymbol{x})/\partial f^{i-1}(\boldsymbol{x}))| = \exp(\mathrm{sum}(s_i(\mathrm{concat}(f^{i-1}(\boldsymbol{x})^{1:d'},c)))) \tag{16}$$

## A.2  Training

As a regeneration learning method, there are two stages during the training. The first stage is to train a graph auto-encoder, PbGMR-GMUS, in this case, through self-supervised learning. This training process happens over all time steps and sequences in the training set. And for a given sequence $[\boldsymbol{Y}_0, \ldots, \boldsymbol{Y}_T]$, we compute $\hat{Y}_t = \mathrm{PbGMUS}(\mathcal{G}, \mathrm{PbGMR}(\mathcal{G}, \mathrm{Y}))$ at each step and minimize the reconstruction loss:

$$\mathcal{L}_{graph} = \sum_{t=1}^{T} \|\boldsymbol{Y}_t - \hat{\boldsymbol{Y}}_t\|_2^2 \quad . \tag{17}$$

In the second stage, the parameters in PbGMR-GMUS will be fixed. And the attention-based sequence model, as well as the conditional flow model, will be trained. We maximize the log-likelihood in the latent space:

$$\mathcal{L}_p = -\frac{1}{T-1} \sum_{t=2}^{T} \log p_\psi(\boldsymbol{z}_t|\boldsymbol{c}_t) \tag{18}$$

We don't set a time window so that the whole trajectory will be trained. The entire training process can be found in Algorithm 1. Particularly in the second stage, such a teacher forcing-like strategy allows a faster training process compared with the previous work [11], which adopts a rollout strategy during the training.

## A.3  Inference

For inference we first encode the first two snapshots into latent space $[\boldsymbol{z}_0, \boldsymbol{z}_1]$. The condition vector $\boldsymbol{c}_t$ is computed from the attention-based sequence model. By sampling a noise vector $\boldsymbol{x}$ from an isotropic Gaussian, and putting it going backward through the flow, we can obtain a new sample from $p_\phi(\boldsymbol{z}_t|\boldsymbol{c}_t)$ as latent representation at each step. We describe the procedure of training in Algorithm 2.

## A.4  Latent analysis

Figure 6 plots the latent $\boldsymbol{z}_t$ and conditional vectors $\boldsymbol{c}_t$ for the stochastic BFS cases. For visualization purposes, we plot the first two modes of the PCA result. Moreover, we plot two different realizations of the stochastic process with clearly distinctive trajectories, as shown in the $\boldsymbol{c}_t$ plot (right). Since the cases are inherently from the same distribution, the sampled latent trajectories $\boldsymbol{z}_t$ plot (left) have a lot of overlaps after the few initial development steps. The plots of different cases visually conform to the physical intuition and indicate that our model can learn the relation between different cases well.

---

**Algorithm 1** Training Process

---

**Input:** Domain graph $\mathcal{G}$, Node features over time $[\boldsymbol{Y}_0, \ldots, \boldsymbol{Y}_T]$, $\text{PbGMR}_\theta$, $\text{PbGMUS}_\phi$, Attention encoder $\text{MHA}_\omega$, Attention decoder Masked-$\text{MHA}_\lambda$, conditional flow model $p_\psi(.|)$, physical condition parameter $\boldsymbol{\mu}$

**Output:** Learned parameters $\theta$, $\phi$, $\omega$ and $\psi$

**repeat**
    **for** $\boldsymbol{Y}_t \in [\boldsymbol{Y}_0, \ldots, \boldsymbol{Y}_T]$ **do**
        $\hat{\boldsymbol{Y}}_t = \text{PbGMUS}_\phi(\text{PbGMR}_\theta(\boldsymbol{Y}_t, \mathcal{G}))$
    **end for**
    Compute $\nabla_{\theta,\phi} \leftarrow \nabla_{\theta,\phi} \mathcal{L}_{graph}(\theta, \phi, [\boldsymbol{Y}_0, \ldots, \boldsymbol{Y}_T], [\hat{\boldsymbol{Y}}_0, \ldots, \hat{\boldsymbol{Y}}_T])$
    Update $\phi$, $\theta$ using the gradients $\nabla_\phi$, $\nabla_\theta$
**until** convergence of the parameters $(\theta, \phi)$       {Self-supervised learning}
$[z_0, z_1, \ldots, z_T] = \text{GMR}_\theta([\boldsymbol{Y}_0, \ldots, \boldsymbol{Y}_T])$
**repeat**
    $\hat{\boldsymbol{\mu}}, \hat{z_0} = \text{MHA}_\omega(\boldsymbol{\mu}, \boldsymbol{z}_0)$
    $[\boldsymbol{c}_2, \ldots, \boldsymbol{c}_t] = \text{Masked-MHA}_\lambda((\boldsymbol{z}_1, \ldots, \boldsymbol{z}_{t-1}), \boldsymbol{I} = (\hat{\boldsymbol{\mu}}, \hat{z_0}))$
    Compute $\nabla_{\omega,\lambda,\psi} \leftarrow \nabla_{\omega,\lambda,\psi} \mathcal{L}_p(\omega, \lambda, \psi, [\boldsymbol{c}_2, \ldots, \boldsymbol{c}_t], [\boldsymbol{z}_2, \ldots, \boldsymbol{z}_t])$
    Update $\omega$, $\lambda$, $\psi$ using the gradients $\nabla_\omega$, $\nabla_\lambda$, $\nabla_\psi$
**until** convergence of the parameters $(\omega, \lambda, \psi)$

---

---

**Algorithm 2** Inference Process

---

**Input:** Domain graph $\mathcal{G}$, The firt two snapshot of node features $[\boldsymbol{Y}_0, \boldsymbol{Y}_1]$, $\text{PbGMR}_\theta$, $\text{PbGMUS}_\phi$, Attention encoder $\text{MHA}_\omega$, Attention decoder Masked-$\text{MHA}_\lambda$, conditional flow model $p_\psi(.|)$, physical condition parameter $\boldsymbol{\mu}$, sample length $T$

**Output:** A trajectory sample $[\boldsymbol{Y}_2, \ldots, \boldsymbol{Y}_T]$
$[\boldsymbol{z}_0, \boldsymbol{z}_1] = \text{PbGMR}_\theta([\boldsymbol{Y}_0, \boldsymbol{Y}_1], \mathcal{G})$
$\hat{\boldsymbol{\mu}}, \hat{z_0} = \text{MHA}_\omega(\boldsymbol{\mu}, \boldsymbol{z}_0)$
**for** $t \in [2, \ldots, T]$ **do**
    $\boldsymbol{c}_t = \text{Masked-MHA}_\lambda((\boldsymbol{z}_1, \ldots, \boldsymbol{z}_{t-1}), \boldsymbol{I} = (\hat{\boldsymbol{\mu}}, \hat{z_0}))[-1]$
    Sample latent representation from conditional flow $\boldsymbol{z}_t \sim p_\phi(\boldsymbol{z}_t | \boldsymbol{c}_t)$
**end for**
Recover physical parameters on the original space $[\boldsymbol{Y}_2, \ldots, \boldsymbol{Y}_T] = \text{PbGMUS}_\phi([\boldsymbol{z}_2, \ldots, \boldsymbol{z}_T])$

---

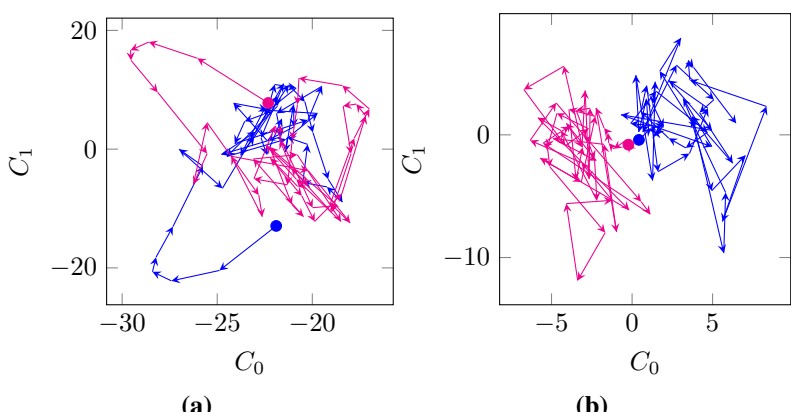

      (a)                 (b)

**Figure 6:** 2-D principle subspace of the (a) latent vectors and (b) conditional latent vectors for the stochastic BFS cases: trajectory 1 (——→), trajectory 2 (——→) start from ——● and ——●, respectively.

### A.5 Dataset

Four fluid simulation datasets, cylinder flow, sonic flow, vascular flow and stochastic backward-facing step (BFS) are used in the experiments. The labeled data are generated by solving the Navier-Stokes equation by OpenFOAM [21], a commonly used finite volume method-based open-source solver for fluid simulation. Cylinder and sonic flow consist of 50 cases in the training dataset and 50 cases in the test dataset. The vascular flow cases have 10 training and 10 test cases. The stochastic BFS case has 5 training cases. Unlike the first three deterministic cases, the stochastic BFS case is a stochastic case. The simulation details can be found in Table 4. Moreover, the details input and output of the PbGMR-GMUS are listed in Table 5.

The governing equation for the first three cases is the same as the one listed in [11]. For the last stochastic BFS case, the governing equation is listed in Equation (19). One significant difference in this case is that it introduces stochasticity by perturbing the inlet $u$ velocity at every time step.

**Table 4:** Simulation details of datasets

| Dataset | Meshing | # nodes | # steps | type |
|---|---|---|---|---|
| Cylinder flow | Fixed | 1699 | 400 | deterministic |
| Sonic flow | Moving | 1900 | 40 | deterministic |
| Vascular flow | Small-varying | 7561 (avg.) | 250 | deterministic |
| Stochastic backward-facing step | Fixed | 22500 | 240 | stochastic |

**Table 5:** Input-output information of PbGMR-GMUS: $u$: X-axis velocity, $v$: Y-axis velocity, $p$: pressure, $T$: temperature, $\rho$: density, $m$: cell volume. $T_{,0}, Re, r, u_{Bi}$ denote the initial temperature, Reynolds number, radius of the thrombus, and boundary velocities, respectively. BFS denotes backward-facing steps.

| Dataset | PbGMR node input | PbGMUS node output | Nodal embed dim | # centers |
|---|---|---|---|---|
| Cylinder flow | $u_i, v_i, p_i, m_i, Re$ | $u_i, v_i, p_i$ | 4 | 256 |
| Sonic flow | $u_i, v_i, p_i, T_i, \rho_i$ $m_i, T_{,0}$ | $u_i, v_i, p_i$ $T_i, \rho_i$ | 4 | 256 |
| Vascular flow | $u_i, v_i, p_i, m_i, r$ | $u_i, v_i, p_i$ | 4 | 400 |
| Stochastic BFS | $u_i, v_i, m_i, u_{Bi}$ | $u_i, v_i$ | 4 | 256 |

$$\frac{\partial u}{\partial x} + \frac{\partial v}{\partial y} = 0,$$
$$\frac{\partial u}{\partial t} + u\frac{\partial u}{\partial x} + v\frac{\partial u}{\partial y} = -\frac{1}{\rho}\frac{\partial p}{\partial x} + \nu(\frac{\partial^2 u}{\partial x^2} + \frac{\partial^2 u}{\partial y^2}),$$
$$\frac{\partial v}{\partial t} + u\frac{\partial v}{\partial x} + v\frac{\partial v}{\partial y} = -\frac{1}{\rho}\frac{\partial p}{\partial y} + \nu(\frac{\partial^2 v}{\partial x^2} + \frac{\partial^2 v}{\partial y^2}),$$

(19)

subject to $u_i^{\text{inlet}} = 10 + u', u' \sim \mathcal{U}(0, 1)$ at every time step.

### A.6 Additional visualization results

Apart from the contour shown in Figure 2, additional contour plots for the cylinder, sonic flow and vascular flow are shown in this section. Figure 7 shows the sampled velocity and std of the velocity for vascular flow cases. Figure 8 presents the sampled pressure for cylinder, sonic and vascular flow. Additionally, Figure 9 shows the sampled temperature variable in sonic flow.

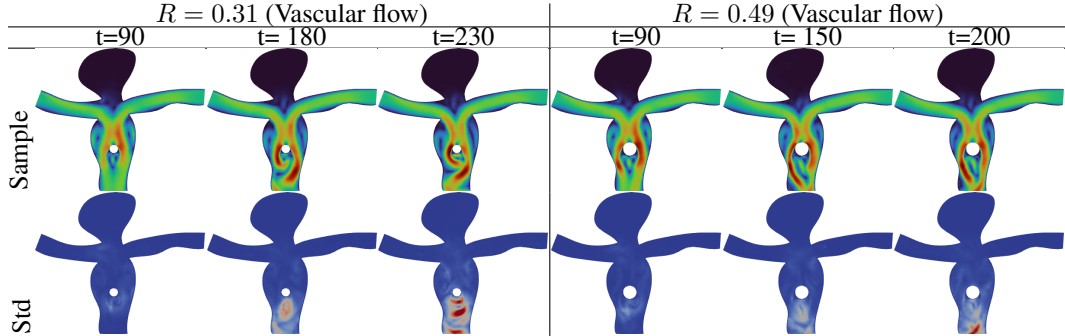

**Figure 7:** Velocity contour for the vascular flow case, the first row is one of our predicted samples. The second row is the predicted standard deviation.

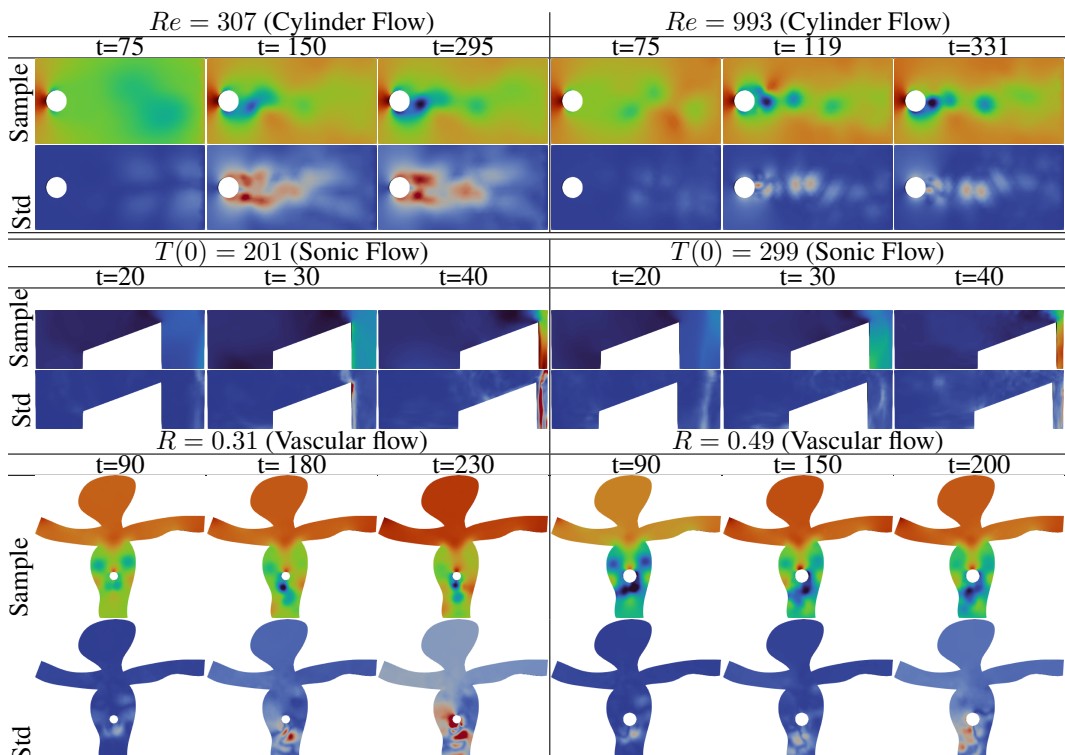

**Figure 8:** Pressure contour for benchmark cases

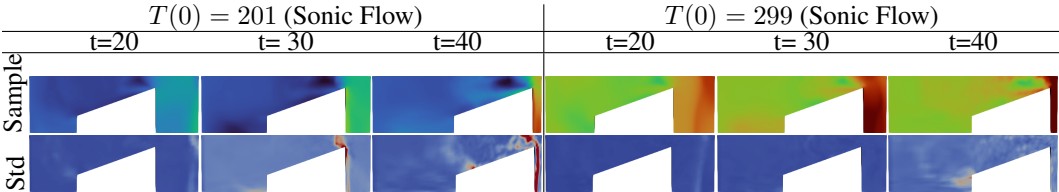

**Figure 9:** Temperature contour for sonic flow case.

### A.7 Multi-head attention

Transformer [29] has been proven successful in the NLP field. The design of the multi-head attention (MHA) layer is based on the attention mechanism with Query-Key-Value (QKV). Given the packed matrix representations of queries $\boldsymbol{Q}$, keys $\boldsymbol{K}$, and values $\boldsymbol{V}$, the scaled dot-product attention used by Transformer is given by:

$$\text{ATTENTION}(\boldsymbol{Q}, \boldsymbol{K}, \boldsymbol{V}) = \text{softmax}\left(\frac{\boldsymbol{Q}\boldsymbol{K}^T}{\sqrt{D_k}}\right)\boldsymbol{V}, \tag{20}$$

where $D_k$ represents the dimensions of queries and keys.

The multi-head attention applies $H$ heads of attention, allowing a model to attend to different types of information.

$$\text{MHA}(\boldsymbol{Q}, \boldsymbol{K}, \boldsymbol{V}) = \text{CONCAT}\left(\text{head}_1, \ldots, \text{head}_H\right)\boldsymbol{W}$$
$$\text{where} \quad \text{head}_i = \text{ATTENTION}\left(\boldsymbol{Q}\boldsymbol{W}_i^Q, \boldsymbol{K}\boldsymbol{W}_i^K, \boldsymbol{V}\boldsymbol{W}_i^V\right), i = 1, \ldots, H. \tag{21}$$

### A.8 Additional details for experimental setups

We described the details of the experiments of PbGMR-GMUS and attention-based conditional flow model. We provide the hyperparameters used in the experiments in Table 6.

| | Cylinder | Sonic | Vascular | Stochastic BFS |
|---|---|---|---|---|
| **PbGMR-GMUS Optimization** | | | | |
| Learning rate | $1 \times 10^{-4}$ | $1 \times 10^{-5}$ | $1 \times 10^{-4}$ | $1 \times 10^{-4}$ |
| Optimizer | | | Adam [51] | |
| Batch size | | | 1 | |
| Number of epochs | 1500 | 20000 | 700 | 600 |
| **MHA/Flow model Optimization** | | | | |
| Learning rate | $1 \times 10^{-4}$ | $1 \times 10^{-5}$ | $1 \times 10^{-5}$ | $1 \times 10^{-4}$ |
| Optimizer | | | Adam | |
| Batch size | 5 | 50 | 10 | 5 |
| Number of epochs | 90000 | 240000 | 240000 | 220000 |
| Weight decay | $1 \times 10^{-5}$ | | N/A | |
| **PbGMR-GMUS Architecture** | | | | |
| Layers of message-passing | | | 3 | |
| Hidden dimension | | | 128 | |
| Output dimension | 256 | 256 | 400 | 256 |
| Activation function | | | relu | |
| **MHA Architecture** | | | | |
| Layers of Encoding MHA | | | 2 | |
| Layers of Decoding Masked-MHA | | | 1 | |
| Hidden dimension | 1024 | 1024 | 1600 | 1024 |
| Activation function | | | gelu [52] | |
| Number of heads | 4 | 8 | 4 | 4 |
| **Flow model Architecture** | | | | |
| Conditioning length | | | 1024 | |
| Hidden size | | | 1024 | |
| Number of coupling layers | | | 2 | |

**Table 6:** Hyperparameters

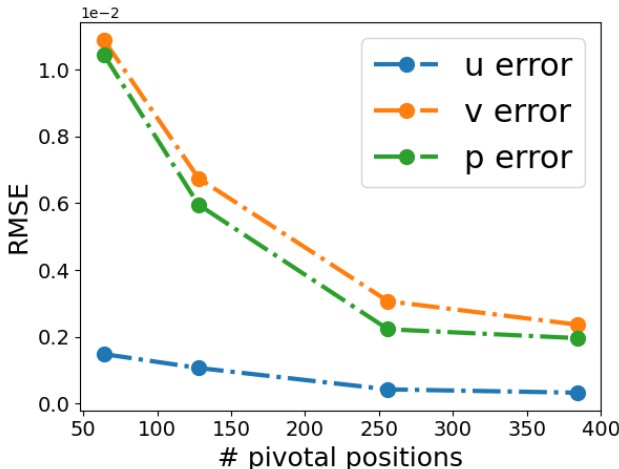

**Figure 10:** Reconstructor error RMSE with different numbers of centers for cylinder flow, with the training epoch $= 300$.

### A.9 Performance impact of substituting Transformer with LSTM

In this research, we conduct an ablation study in which we replace the proposed attention-based sequence model in our framework with a Long Short-Term Memory (LSTM) [53] architecture. The aim is to evaluate the impact of the underlying sequence modeling on the performance of our system. The result is listed in Table 7. Following this change, we observe that there is a notable decrease in the performance metrics, suggesting a less optimal fit for the task at hand compared to the Transformer-based model. This outcome underscores the usefulness of attention-based sequence models, like the Transformer, which appears to capture dependencies in the data more effectively than the LSTM. The attention mechanism inherent to Transformers allows the model to focus on different parts of the input sequence dynamically, which may explain the observed performance superiority.

**Table 7:** The average relative rollout error of two systems, with the unit of $1 \times 10^{-3}$.

| Dataset-rollout step | | Sonic flow-40 | | | | Vascular flow-250 | | |
|---|---|---|---|---|---|---|---|---|
| Variable | | $u$ | $v$ | $p$ | $T$ | $u$ | $v$ | $p$ |
| PbGMR-GMUS | LSTM | 10.3 | 29.7 | 1.7 | 11.1 | 43.0 | 46.6 | 16.6 |

### A.10 Performance impact of number of centers

We conduct an ablation study to investigate the effect of the number of centers on the RMSE of our framework's predictions for cylinder flow recovery. The result is in Figure 10. As we progressively increased the number of centers, the recovery RMSE exhibited a decreasing trend, demonstrating an enhancement in prediction accuracy. Interestingly, upon reaching 256 centers, the rate of RMSE decrease started to decelerate significantly. This suggests that the optimal number of centers for cylinder flow is around 256. Beyond this point, further increments do not contribute as much to the improvement in encoding-decoding accuracy, while simultaneously increasing the computational cost. Hence, a count of 256 centers appears to be the sweet spot, offering a good trade-off between recovery accuracy and computational efficiency.

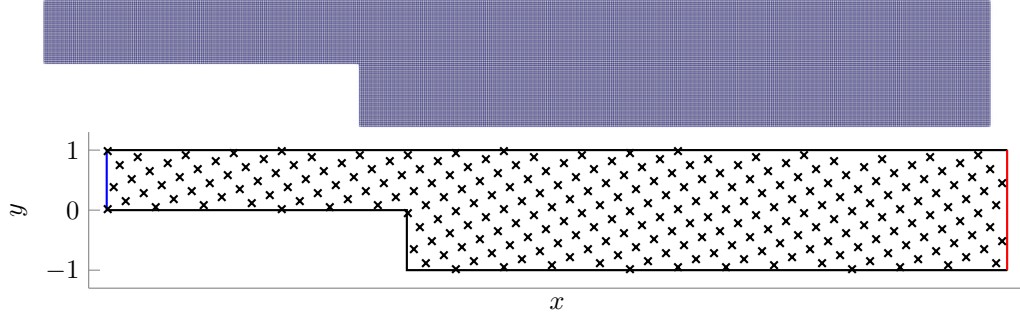

**Figure 11:** The first row shows meshes and the second row shows the distribution for the centers.

## A.11 Centers

The centers distribution for Stochastic BFS can be found in Figure 11.

## A.12 Analysis for stochastic BFS

Backward-facing step flow refers to a fluid flow configuration characterized by a sudden contraction followed by an expansion in a channel or pipe. This flow configuration is commonly encountered in various engineering applications, such as in heat exchangers, combustion chambers, and aerodynamic systems. In backward-facing step flow, the fluid initially enters the channel or pipe through an inlet, where it encounters a step or abrupt contraction. As the fluid passes over the step, its velocity increases, and the flow separates, forming a recirculation zone downstream. This recirculation zone is often referred to as the "backflow region." The backward-facing step flow has been extensively studied due to its complex flow characteristics and its relevance in understanding phenomena like separation, reattachment, and flow control.

The simulations and generated samples exhibit remarkable similarity in the contours of both stream-wise velocity and wall-normal velocity (Figure 4, 12, 13, 14). This close resemblance demonstrates the high fidelity of the simulation results and the accuracy of the generated samples. The level of detail captured in these contours highlights the effectiveness of the proposed generative framework.

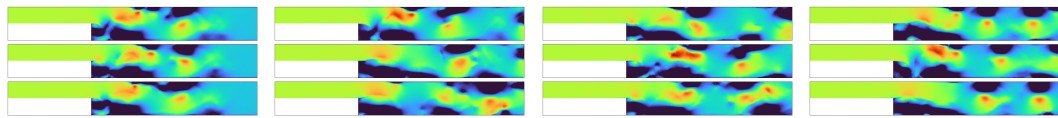

**Figure 12:** Velocity contour ($v$) at time step $40, 80, 120, 160$ (*left to right*) from different samples.

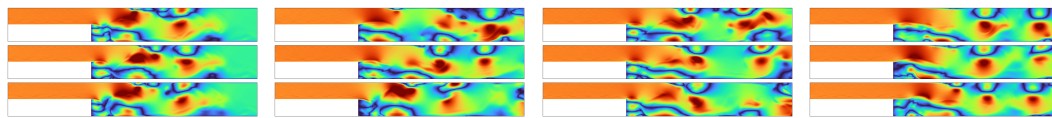

**Figure 13:** Velocity contour ($u$) at time step $40, 80, 120, 160$ (*left to right*) from different LES simulations.

We also randomly choose nine spatial locations to analyze the learning performance, and the coordinates of these points are plotted in Figure 15. Figure 16 and 17 presented the comparison of $u$, $v$ velocity between the LES simulation, URANS simulation and our model. Generally, the pointwise velocity temporal signals obtained from the generated sample provide a comprehensive representation of the flow characteristics and look visually similar to those obtained from LES simulations. However, the traditional URANS modeling approach cannot capture the intricate fluctuations in the flow field. Note that we also found the generated velocity time signals at points 1 and 2 have a relatively large discrepancy from the LES simulation, and it is explainable. Point 1 is very close to the inlet boundary; therefore, the $u$ velocity here is very random because of our stochastic velocity inlet condition. Point 2 locates in a region where the BFS flow is not fully developed yet and in this region the $v$ velocity

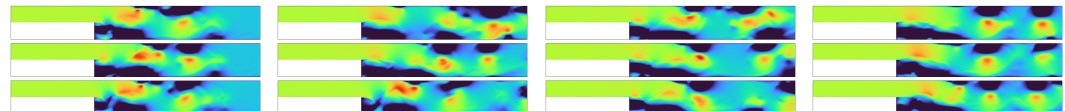

**Figure 14:** Velocity contour ($v$) at time step $40, 80, 120, 160$ (*left to right*) from different LES simulations.

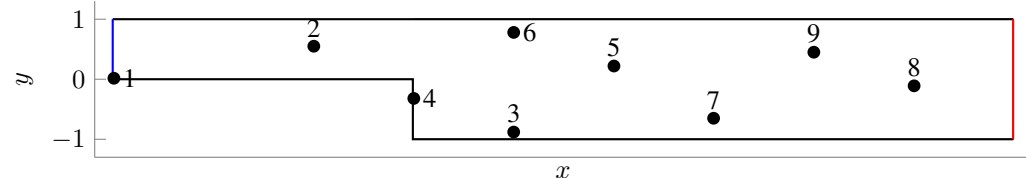

**Figure 15:** Spatial location of the 9 random chosen evaluation points. Point 1: $(-4.98, 0.016)$, Point 2: $(-1.65, 0.55)$, Point 3: $(1.68, -0.88)$, Point 4: $(0.016, -0.32)$, Point 5: $(3.35, 0.22)$, Point 6: $(1.68, 0.78)$, Point 7: $(5.01, -0.65)$, Point 8: $(8.35, -0.11)$, Point 9: $(6.68, 0.45)$

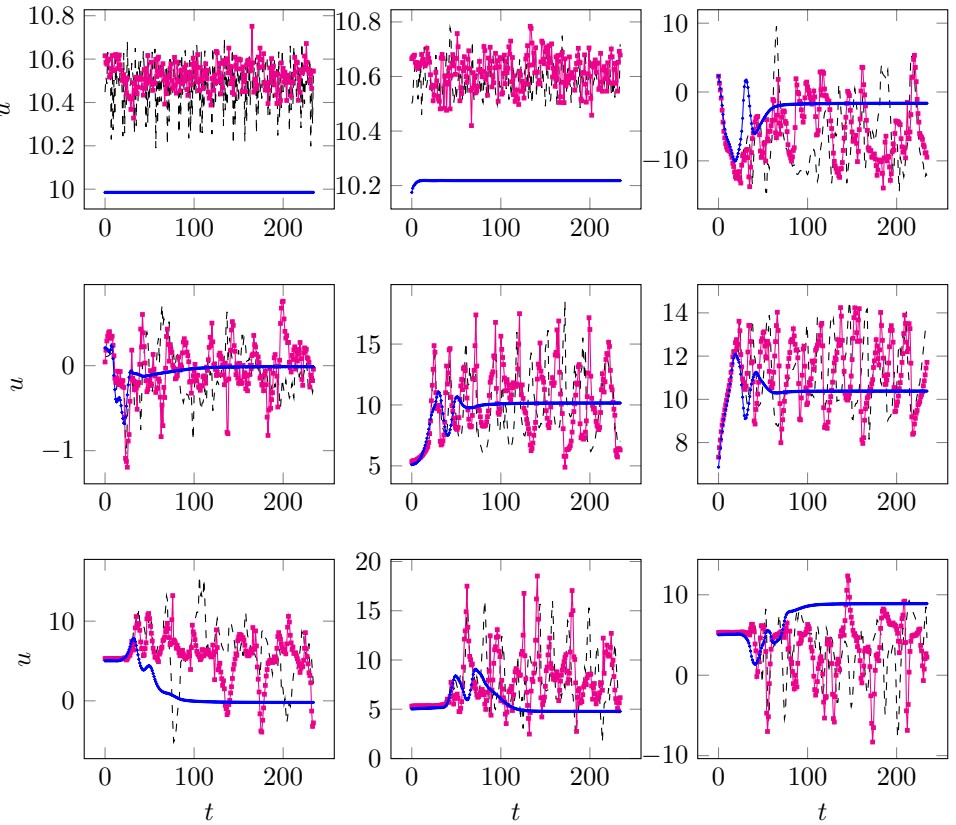

**Figure 16:** Time signal from LES (- - -), URANS (——), flow model (—■—) of $u$ evaluated at 9 points (*top to bottom, left to right*).

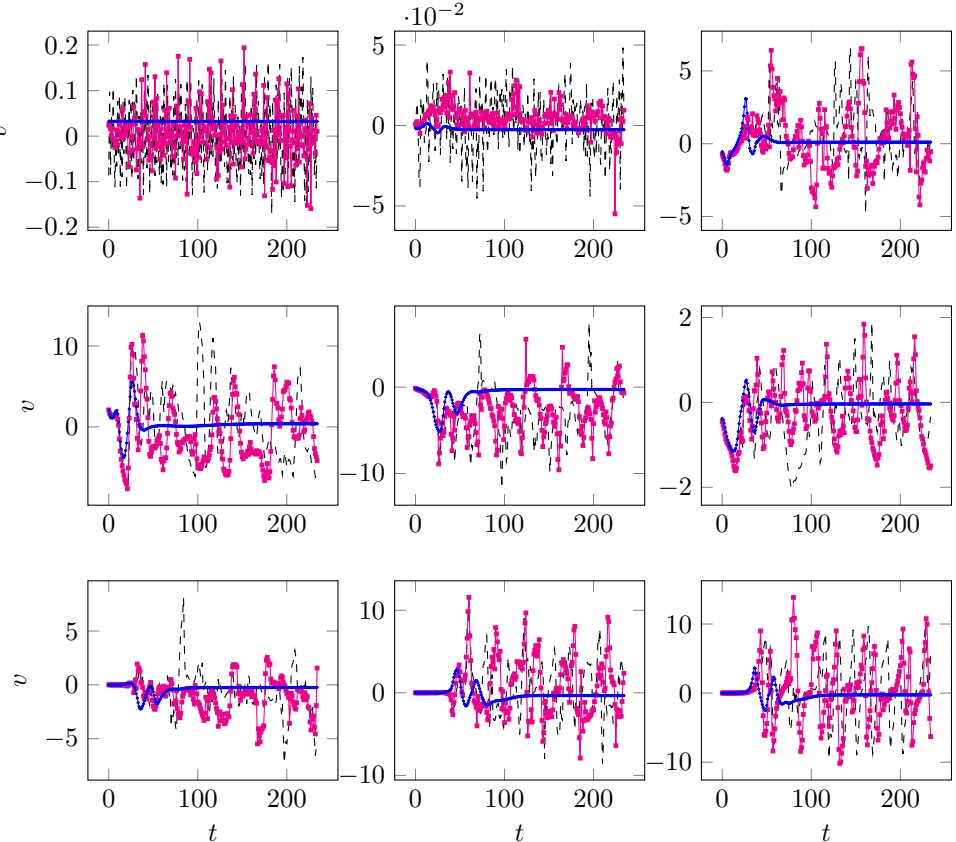

**Figure 17:** Time signal from LES (- - -), URANS (——), flow model (—■—) of $v$ evaluated at 9 points (*top to bottom, left to right*).

magnitude is very small. These conditions make it hard to capture an accurate distribution of the velocity at these 2 points. Finally, we also compare the velocity distribution between our generated samples and the LES simulation results, as shown in Figure 18, 19. Except for points 1 and 2, the distribution of our generated samples aligns very well with the LES simulations.

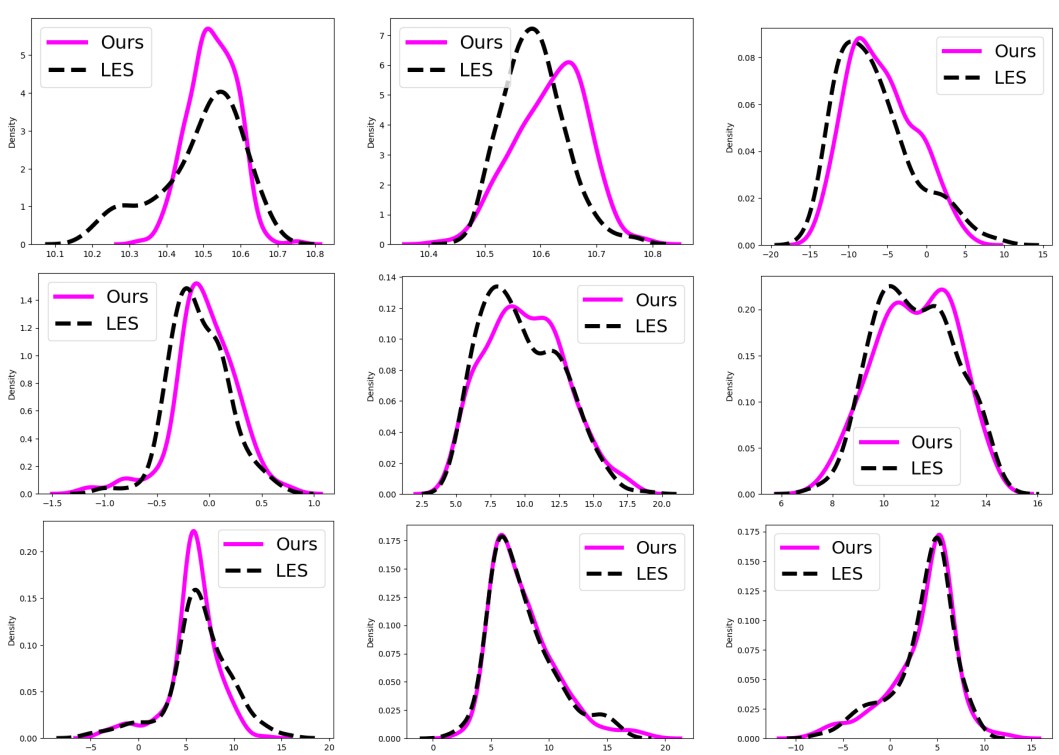

**Figure 18:** Comparison of the distribution of $u$ velocity between the LES result and our model at 9 evaluation points

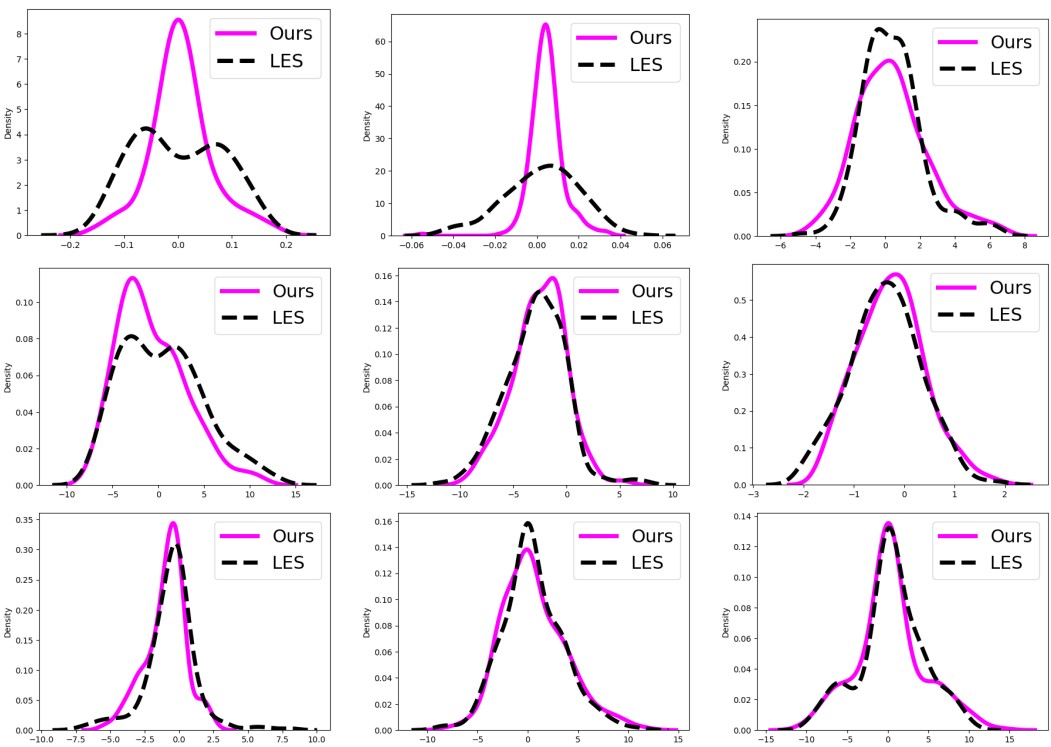

**Figure 19:** Comparison of the distribution of $v$ velocity between the LES result and our model at 9 evaluation points

## A.13 Ablation study

Comprehensive ablation studies are included in this section to demonstrate the effectiveness of each parts. The velocity reconstruction error on the Backward-facing step flow dataset with or without residual connection and centers is shown in Table 8. The result indicates that each component can improve the graph reconstruction accuracy. We also test the influence of PbGMR-GMUS and the attention conditional flow model to the final performance of the deterministic task, as shown in Table 9. The result shows that both the encoder-decoder part and the flow model are necessary for the success of the whole framework.

**Table 8:** The reconstruction error of stochastic flow with the same training epoch number for Backward-facing step flow dataset. The residual connection and centers can decrease the reconstruction error significantly.

| residual connection | centers | $v$ |
|:---:|:---:|:---:|
| ✗ | ✗ | 85 |
| ✗ | ✓ | 19.2 |
| ✓ | ✓ | **15.8** |

**Table 9:** Ablation study: the average relative rollout error with/without each proposed component for vascular flow. CNF: using conditional normalizing flows for sequence prediction. In variant 2, we remove the CNF block and use a transformer only structure for predictions, similar to [11]. Our model with residual connection and centers have the best performance.

| | residual connection+centers | CNF | $u$ | $v$ | $p$ |
|:---:|:---:|:---:|:---:|:---:|:---:|
| GMR-GMUS | ✗ | ✗ | 7.3 | 10 | 22 |
| Variaint1 | ✗ | ✓ | 6.1 | 8.7 | 9.2 |
| Variaint2 | ✓ | ✗ | 5 | 8 | 3 |
| Ours | ✓ | ✓ | **3.49** | **5.47** | **1.05** |

To efficiently incorporate the temporal information, we conceive a temporal conditional normalizing flows to account for the long-time temporal dependency. The original RealNVP is only a probabilistic model, and it is not straighforward to include temporal information. To achieve this, we propose an encoding-decoding transformer structure to incorporate fixed physical parameters and all previous steps into condition vector $c$. In this ablation study, we remove the transformer structures and replace condition vector $c$ by the current step latent vector $z_{t-1}$ when predicts next step $z_t$. This variant can be treated as a probabilistic 1-step prediction method. The result in Table 10 shows that the proposed structure can significantly improve the accuracy compared with such 1-step method, while still easy to calculate the probability of a coupling layer if we concatenate condition vector at every coupling layer, as shown in Appendix A.1.

**Table 10:** Ablation study: effect of attention-based temporal conditional model for Sonic dataset, NF: normalizing flows only condition on previous step. The proposed model outperforms 1 step NF, indicating the design of attention based temporal model is necessary for accurate time series prediction.

| | $u$ | $v$ | $p$ | $T$ |
|:---:|:---:|:---:|:---:|:---:|
| 1-step NF | 4.85 | 16.36 | 1.09 | 5.50 |
| Ours | **0.37** | **0.85** | **0.079** | **0.01** |

Our model can benefit from longer time dependencies, since we take all previous steps in the experiment setting. Also, we can use the moving window to reduce the cost further. We did one additional experiment of reducing the number of inputs during the inference time, as shown in Table 11. When reducing the window size from 400 to 150, the degradation in accuracy is not significant. So the model is also applicable to cases with thousands or even longer steps with the sliding window.

**Table 11:** Ablation study: effect of window length (WL) for Cylinder dataset, reducing the window length to 150 doesn't increase error significantly.

| WL | $u$ | $v$ | $p$ |
|---|---|---|---|
| 150 | 4.14 | 80.48 | 22.99 |
| 400 | **3.8** | **74** | **20** |

Though we only tested RealNVP in the experiment part, other normalizing flows are also feasible here. In Table 12, we test another normalizing flow model MAF [54], and find the result is comparable to RealNVP. It indicates that the proposed framework is flexible.

**Table 12:** Ablation Study: the average rollout error using different normalizing flow models. MAF: Masked Autoregressive Flow. The performance is comparable with RealNVP.

| Dataset-rollout step | | Cylinder flow-400 | | | Sonic flow-40 | | | Vascular flow-250 | | | |
|---|---|---|---|---|---|---|---|---|---|---|---|
| Variable | | $u$ | $v$ | $p$ | $u$ | $v$ | $p$ | $T$ | $u$ | $v$ | $p$ |
| Ours | RealNVP | **3.8** | 74 | 20 | 0.37 | 0.85 | 0.079 | **0.01** | **3.49** | **5.47** | **1.05** |
| Ours | MAF | **3.8** | **72** | **19.13** | **0.33** | **0.69** | **0.055** | 0.02 | 3.82 | 5.84 | 1.16 |

As a probabilistic model, the proposed PbGMR-GMUS + conditional flow model is trying to fit such a stochastic process and get various samples with the same physical statistics. To demonstrate these, we test MGN and GMR-GMUS on the stochastic dataset. In Table 13, we can see that the performance of the proposed model is better than other learning-based models on all metrics. In Figure 20, we visualize the mean and variance of velocity from each model, and find the MGN can not capture the real distribution at all. The result of GMR-GMUS model, though looks reasonable in statistics, can only produce the same output given the same input condition, which doesn't reflect the stochasticity of the last dataset. In Figure 21, we get two samples from GMR-GMUS, and find the two samples are exactly the same.

**Table 13:** Criteria of generation quality for the stochastic flow for URANS and different learning-based models, the proposed model has the best performance.

| | CRPS ($C_u$,↓) | CRPS ($C_v$,↓) | FVD ($d_u$,↓) | FVD ($d_v$,↓) | MFE ($e_u$,↓) | MFE ($e_v$,↓) | TE (TKE,↑) |
|---|---|---|---|---|---|---|---|
| URANS | 3.24 | 2.0 | 228112 | 137860 | 0.31 | 0.94 | 0.192 |
| MeshGraphNet | 5.06 | 2.47 | 1159950 | 179206 | 0.82 | 1.60 | 0.87 |
| GMR-GMUS | 2.57 | 2.27 | 4976 | 3740 | 0.167 | 1.96 | 0.99 |
| Ours | **1.28** | **1.08** | **1262** | **397** | **0.0176** | **0.176** | **0.99** |

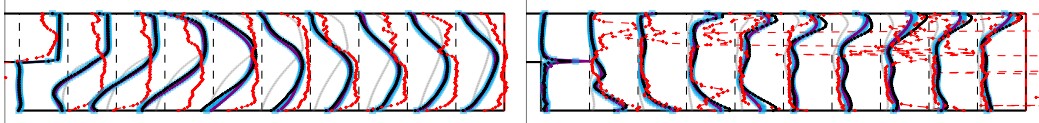

**Figure 20:** Evaluation lines of **mean** (*left*) and **variance** (*right*) of streamwise velocity $u$: $x$-evaluation lines (- - -). Quantities from LES (——), URANS (——), MeshGraphNet (- - -), GMR-GMUS (——) and the proposed model (——)

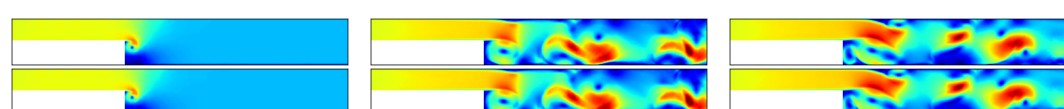

**Figure 21:** Velocity magnitude at 3 time steps $0, 50, 100$ (*left to right*) from different samples using GMR-GMUS + transformer for the stochastic dataset. 1st row for sample 1. 2nd row for sample 2. The deterministic model can only produce the same output while the proposed model can generate different realizations.

