# OpenReview forum: "Unifying Predictions of Deterministic and Stochastic Physics in Mesh-reduced Space with Sequential Flow Generative Model"
_NeurIPS.cc/2023/Conference — NeurIPS 2023 spotlight_

### Official Review · Reviewer_8v2N · 2023-06-26

**Soundness:** 4 excellent
**Presentation:** 3 good
**Contribution:** 3 good
**Rating:** 6
**Confidence:** 4

**Summary:**

The paper proposes PbGMR-GMUS to encode the graphs of physics systems into low-dimensional features, which are further reconstructed into the desire graph representations. An attention-based model is integrated into flow-based generative models to predict the future dynamics in the latent space. The proposed method is able to handle both deterministic and stochastic fluid dynamics thanks to the probabilistic model and achieve superior performance over previous methods.

**Strengths:**

* An effective model to learn the graph representations in low-dimensional space.
* The simulation of dynamics is achieved by a probabilistic method using generative models, which rollout vivid predictions for long-term predictions.
* The method is able to solve both deterministic and stochastic fluid dynamics.

**Weaknesses:**

* Miss citation at L62 for GMR-GMUS.
* On top right of Figure 1, is it "Coupling layer" instead of "Couling layer"?
* Unclear caption for Figure 5. Which one is the variance?


**Questions:**

* How to prove the statement about the first benefits at L163-164? Is there any further explanation?
* Is the main difference to between simulating deterministic and stochastic fluid dynamics that the $\mu$ in Equation 14 would change with time for stochastic dynamics?
* Any video demos for the predictions of both deterministic and stochastic fluid dynamics? The temporal consistency can be better illustrated by videos.



**Limitations:**

The proposed PbGMR-GMUS is technically limited, which is a variant of GMR-GMUS with simple modifications. Video demos are missing to verify the temporal consistency of the predictions.

---

> ### Author Rebuttal · Authors · 2023-08-10
>
> We appreciate your important support. And we seriously address your concerns accordingly; they are extremely helpful to the submission.
>
> > **Q1:** Miss citation at L62 for GMR-GMUS.
>
> **Response:** Thanks for pointing it out. We already added reference in the revised version.
>
> > **Q2:** On top right of Figure 1, is it "Coupling layer" instead of "Couling layer"?
>
> **Response:** Yes, you are right. It should be "coupling  layer" and we already fixed it.
>
> > **Q3:** Unclear caption for Figure 5. Which one is the variance?
>
> **Response:** We agree that the caption should be more clear. The upper is the mean of the velocity, and the bottom is the variance of the velocity. We added more explanations in the revised version.
>
>
> > **Q4:** How to prove the statement about the first benefits at L163-164? Is there any further explanation?
>
> **Response:** As a stochastic process, probabilistic models on the original graph space $p(Y_t|Y_{t+1})$ face severe one-to-many mapping problems (ill-posed or ill-condition problem)[24]. That is, given the previous step state $Y_t$, it can correspond to many different next step $Y_{t+1}$, which increases the learning difficulty. Incorrect modeling on the
> ill-posed problem could result in overfitting on the training set and poor generalization on the test set. Thus, converting $Y$ into more compact $z$ will make the mapping between $z_t$ and $z_{t+1}$ less ill-posed and ease the model learning. We will add more explanation in the revised version.
>
> > **Q5:** Is the main difference to between simulating deterministic and stochastic fluid dynamics that the $\mu$ in Equation 14 would change with time for stochastic dynamics?
>
> **Response:** Thanks for asking the clarification. Here $\mu$ is a time-invariant global physical parameter, such as $Re$. And the numerical simulation has another time-varying parameter such as the stochastic boundary condition for the stochastic fluid dynamics. As for our model, predictions of deterministic and stochastic physics are unified; the model can automatically capture the stochasticity conditioning on $\mu$ during the training.
>
> >  **Q6:** Any video demos for the predictions of both deterministic and stochastic fluid dynamics? The temporal consistency can be better illustrated by videos.
>
> **Response:** Thanks for this good suggestion. We already uploaded some video demos. This link is anonymized and contains not author information. Based on the policy, we first sent it to the AC in the official comment.
>
> We appreciate your careful reading and suggestions for this paper. We believe these revisions and new supplementary experiments help improve the manuscript a lot. And your support means a lot to us.

---

### Official Review · Reviewer_9WF1 · 2023-07-05

**Soundness:** 3 good
**Presentation:** 2 fair
**Contribution:** 3 good
**Rating:** 5
**Confidence:** 4

**Summary:**

This paper introduces a novel mesh-based machine-learning approach for modeling stochastic fluid dynamics. Similar to [11], the state transitions are modeled in a compact latent space (referred to as the mesh-reduced space) rather than the high-dimensional mesh space. Compared with [11], this paper makes several technical contributions below.
1. It proposes to use virtual pivotal positions instead of selecting pivotal nodes from the mesh topology.
2. It employs a deep generative model based on a normalizing flow approach, RealNVP, to model the stochastic dynamics. The model also incorporates multi-head attention over sequential latent states.

The experimental results demonstrate that the proposed model significantly outperforms [11] in the deterministic setup, while in the stochastic setup, the model is compared against an off-the-shelf CFD method named URANS, rather than learning-based models.

**Strengths:**

Originality: To the best of my knowledge, this paper introduces the first learning-based probabilistic model that simulates fluid systems upon the mesh topology. The key innovation lies in integrating a flow method for stochastic temporal modeling into the GMR-GMUS [11] framework, which presents a reasonable and creative combination of existing approaches.

Quality and clarity: Overall, the paper is clear and easy to follow.

Significance: The proposed model is shown to be effective in deterministic setups, which consistently outperforms MeshGraphNet and GMR-GMUS [11] by substantial margins.

**Weaknesses:**

Main concerns:

1. The proposed model appears to be a straightforward extension of GMR-GMUS [11] combined with RealNVP [25].
2. Ablation studies for using virtual pivotal positions, residual connections, and the conditional normalizing flow are lacking, which makes it difficult to determine the specific contributions of these components to the performance improvement shown in Table 1. It is suggested that the authors address this issue and provide a more comprehensive analysis.
3. Although the paper addresses the new problem of probabilistic modeling of fluids, the lack of comparison with existing learning-based models such as GMR-GMUS and MeshGraphNet in Section 5.2 raises doubts about the true challenges of the prediction task for stochastic fluid systems and the superiority of the proposed approach.

Minor concerns:

1. Accurate citations with corresponding publication sources should be provided. For example, GMR-GMUS [11] was published at ICLR 2022. his will enhance the credibility and traceability of the referenced papers.
2. It would be valuable to include a brief discussion regarding the limitations of the research to provide readers with a broader perspective on the potential areas for improvement or expansion.
3. In Line 62, the authors introduce GMR-GMUS without relevant sources, which made me confused until I came to Section 3.1.
4. In Lines 113-121, it would be good if the authors could describe the learning method by referring to specific components or steps in Figure 1, so the readers can better grasp the concepts and processes discussed in the text.

**Questions:**

1. In Line 118, the claim that the proposed model enables the enlargement of the detection distance compared to GMR-GMUS lacks detailed explanations. Further elaboration on this point would be beneficial.
2. In Line 87, it is mentioned that the stochastic dynamic system includes a random parameter $\mu$ that varies over time, such as perturbations in the boundary conditions of turbulent flow. However, in Line 161, $\mu$ is referred to as the global physical system parameter and seems to be time-invariant. It would be helpful for the authors to clarify whether $\mu$ is time-varying across the input and output sequences.
3. Regarding $\mu$, as well as the positions $p(i)$ of the mesh cells, it is not clear from the provided context whether they are given in the future prediction time horizon.
4. I am not very familiar with RealNVP, but based on the information provided, should it be written as $P(z_{1:T} \mid \mu, z_0, x; \theta)$ in Equation (10) instead of $P(z_{1:T} \mid \mu, z_0; \theta)$?





**Limitations:**

No discussions on limitations or broader societal impacts are presented in the current text.

---

> ### Author Rebuttal · Authors · 2023-08-10
>
> First and foremost, we express our heartfelt gratitude for your warm acknowledgment of the novelty of the learning-based probabilistic model, meticulously simulating fluid systems across intricate mesh topologies. Here are our responses to your questions.
>
> > **Q1:** The proposed model appears to be a straightforward extension of GMR-GMUS [11] combined with RealNVP [25].
>
> **Response:**  Our endeavors transcend the boundaries of the deterministic system set by the prior SOTA works, significantly improved the SOTA reconstruction technique, and more importantly, without loss of the generalizability, unified the deterministic and stochastic system.
>
> By replacing pivotal nodes with pivotal positions, we can enlarge the detection distance for each position embedding, thus aggregating information more efficiently. Also, a residual connection is added to the message-passing layer, which can make the training more stable. Such advantages make PbGMR-GMUS's reconstruction accuracy higher against GMR-GMUS. And low reconstruction error is significant for the success of latent generative models. To test the improvement brought by PbGMR-GMUS, we conducted comprehensive ablation studies, and the results are shown in Tables 8 and 9 (supplementary experiments). The result shows that all the newly added components can improve the performance. Detailed analysis is shown in the global rebuttal.
>
> Furthermore, our attention-based temporal conditioned generative model is not a simple extension of RealNVP. The original RealNVP is only a probabilistic model, and it is not straighforward to include temporal information. To achieve this, we propose an encoding-decoding transformer structure to incorporate fixed physical parameters and all previous steps into condition vector $c$. Then we prove that it is still easy to calculate the probability of a coupling layer if we concatenate condition vector $c$  at every coupling layer. Actually, the key design is attention-based temporal conditional model, not the selection of RealNVP. The proposed framework can be combined with other probabilistic model. To prove this, we replace RealNVP with MAF, another flow model, Table 12 indicates the results are comparable to RealNVP. So the whole framework is a flexible and useful paradigm. We also test within a single RealNVP, removing the temporal conditional model. The performance in Table 10 drops very obviously.
>
>
> > **Q2:** Ablation studies for using virtual pivotal positions, residual connections, and the conditional normalizing flow are lacking ...
>
> **Response:** We obey your command. We have added the ablation studies on virtual pivotal positions, residual connections, and conditional normalizing flows and the results are in Table 8-10 (supplementary experiment results). In general, all of the proposed components help improve the performance. Detailed analyses are in global rebuttal **Q1**.
>
> > **Q3:** The lack of comparison with existing learning-based models such as GMR-GMUS and MeshGraphNet for stochastic fluid systems ...
>
> **Response:** The reviewer's concern is absolutely reasonable. We agree that more learning-based baselines are needed for section 5.2 and train the GMR-GMUS[11] and MeshGraphNet[3] to address the concern. The results are attached in Table 13 and Figure 20. Our model has the best performance, and the deterministic model cannot capture the critical stochastic nature of the underlying physics. Detailed analyses are shown in the global rebuttal **Q2** due to the word limitation.
>
> > **Q4:** ... clarify whether $\mu$ is time-varying across the input and output sequences.
>
> **Response:** Thanks for your help with clarification. The correct form for Equation 2 is $\frac{\partial{u}}{\partial{t}} = j(\mathbf{u}, \mathbf{\mu}, \mathbf{\iota})$ where the $\iota$ is the random parameter over time for stochastic systems (e.g., boundary conditions). So $\mu$ in lines 71, 86, and 87 should be replaced with $\iota$.
> In Line 161, the $\mu$ is the global physical system parameter and is time-invariant (e.g., Re number). We revised the notations to make it more clear. We appreciate this excellent comment.
>
> > **Q5:** Regarding $\mu$, as well as the positions $p(i)$ of the mesh cells, it is not clear from the provided context whether they are given in the future prediction time horizon.
>
> **Response:** Thank you for asking the clarification. During time series prediction in the latent space, $\mu$ (time-invariant global physical parameter) is given through the transformer encoder in equation (12). While the positions of the mesh cells are not given during the prediction in time. We only need to guarantee that the node order is fixed when concatenating them into latent vector $z$.
>
> >  **Q6:** Should it be written as ... in Equation (10) instead of ...?
>
> **Response:** Thanks for raising this discussion. RealNVP is a probabilistic model. We define such conditional probability only on latent space. It should be noticed that here 'latent space' is not the same as mentioned in section 2.2, which describes internal operation of RealNVP. In Equation 10, we only care about how to define such temporal conditional probability, and there is no relation with specific probabilistic model. Though we select normalizing flow model in this work, it can be defined by other conditional distributions.
>
> > **Q7:** It would be valuable to include a brief discussion regarding the limitations of the research.
>
> **Response:** Thanks for this concern. We agree it is necessary to discuss the limitation of the proposed model. Please refer to global rebuttal **Q3** for details.
>
> > **Q8:** Accurate source of GMR-GMUS [11], and proper reference for GMR-GMUS in Line 62.
>
> **Response:** Thanks for pointing it out. We revised several references with accurate sources and added a reference in Line 62.
>
> We believe your suggestions are helpful. Thanks a lot for your careful reading and efforts on our submission. We revised the minors in our manuscript.

---

> > ### Author Response · Authors · 2023-08-12
> > **Additional reply**
> >
> > Because of the word number limitation, there are two questions we can not reply to in the rebuttal. But we think these two suggestions are very important and we already revised our manuscript based on these questions. We add our replies to these questions here:
> >
> > > **Q9:** In Line 118, the claim that the proposed model enables the enlargement of the detection distance compared to GMR-GMUS lacks detailed explanations. Further elaboration on this point would be beneficial.
> >
> > **Response:** We agree that we should add detailed explanations here. For pivotal nodes, it can only obtain nearby nodes through message-passing layers. However, due to over-smoothing issues, the number of message-passing layers can not be large (generally, people adopt 4 or 5 layers). Thus, pivotal nodes can only detect nodes within graph distance 4 or 5. Therefore, the real spatial distance is very small. However, for the pivotal position in equation (8), it selects nodes directly based on spatial distance. Thus, choosing a relatively large $k$ can consider more nodes and detect a larger area at the beginning of recovery.
> >
> > > **Q10:** In Lines 113-121, it would be good if the authors could describe the learning method by referring to specific components or steps in Figure 1, so the readers can better grasp the concepts and processes discussed in the text.
> >
> > **Response:** Thanks for this perceptive suggestion. We believe it will be really helpful for readers to understand the whole framework. We followed this suggestion in our revised version.

---

> > ### Comment · Reviewer_9WF1 · 2023-08-18
> >
> > Thank you for the replies to my questions and comments. After reading the other reviews and answers, most of my concerns are addressed, so I’d be happy to support acceptance, and I’ll raise my score.

---

> > > ### Author Response · Authors · 2023-08-18
> > > **Thank you so much for your support**
> > >
> > > Thank you so much for your time and efforts in reviewing this paper. Your insightful questions really help us improve the manuscript. We really appreciate that you support acceptance of our submission!

---

### Official Review · Reviewer_YdeK · 2023-07-05

**Soundness:** 4 excellent
**Presentation:** 3 good
**Contribution:** 3 good
**Rating:** 8
**Confidence:** 3

**Summary:**

The authors present a unified framework for solving deterministic and stochastic physical dynamical systems on high-dimensional mesh space. Instead of updating values at each discretized mesh, the paper introduces an approach that evolves states in a low dimensional latent space through encoding states and physical parameters. This encoding is done by a message passing graph neural network and multi-head attention (MHA) model, which encodes physics parameters into a conditional vector. The encoded state is then evolved using a conditional normalizing flow, dependent on the encoded conditional vector. The effectiveness of this method is shown through experiments, in which it outperforms other mesh-based ML models in terms of accumulation error in deterministic problems, with applications in stochastic problems also being demonstrated.

**Strengths:**

The proposed method is simple and applicable to wide range of PDE-simulation problems with discretized domain. Wide variety of experiments are conducted, and both quantitative and qualitative evaluations are provided.

**Weaknesses:**

About the novelty, it is still unclear that what components of the proposed model enables solving both deterministic and stochastic  systems efficiently. Are there any limitations that prevent existing models from being applied to both systems?

It is concerning that the proposed method take all the previous (latent) vectors as input to predict the next state, as opposed to fixed small number of input vectors of MeshGraphNets reported in Section 5, that may result in unfair advantages over other methods. Also, although in Table 1 the reconstruction error of PbGMR-GMUS are compared against GMR-GMUS, it is still unclear how the proposed model would compare against an ablation model whose encoder is GMR-GMUS.

The proposed model is compared against baselines which evolve states in original space. There is a possibly missing reference [1], and the motivation of using latent evolution is similar to the presented one. How does this work compare against it? Is the proposed method also applicable to grid space?

[1] Tailin Wu, Takashi Maruyama, and Jure Leskovec. Learning to accelerate partial differential equations via latent global evolution (NeurIPS 2022)


**Minor comments**

In Appendix A.10, Figure 11 is cited, but seemingly wrong.

**Questions:**

Please have a look at the weakness mentioned in the strength and weakness section and address these. Overall the idea seems interesting, the authors need to substantiate their claims in light of existing literature and possibly more experiments.


**Limitations:**

Yes, the limitations are discussed in the appendix.

---

> ### Author Rebuttal · Authors · 2023-08-10
>
> The reviewer is positive about the application and performance to wide range of PDE-simulation problems of the proposed framework. This is an excellent support to our work. And reviewer encourages us to further experiment to highlight our novelty over previous models. Here are our responses to your questions.
>
> > **Q1:** It is still unclear that what components of the proposed model enable solving both deterministic and stochastic systems efficiently. Are there any limitations that prevent existing models from being applied to both systems?
>
> **Response:** This is a great and important question. Please refer to our global rebuttal **Q2**. The previous models' most challenging part is predicting stochastic systems in mesh space. Our new experiment results indicate previous SOTA models can not capture close distribution or get different samples with the same physical statistics. (Table 13, Figure 20, and Figure 21 in new supplementary experiments)
>
> Also, existing CNN-based models, such as video generation models, can not handle irregular mesh space. To overcome the above limitations, we first propose PbGMR-GMUS to encode the graph space into latent space with highly accurate reconstruction. A novel attention-based temporal conditioned flow model is then developed as the probabilistic model. At each time step, only a low-dimensional vector needs to be predicted. And the model can capture spatiotemporal dependencies and stochastics efficiently.
>
>  We did comprehensive ablation studies, as shown in Table 8 and Table 9 in the one-page rebuttal. The results demonstrate that the proposed model can increase the accuracy of deterministic training. Moreover, the proposed flows model enables learning stochastic processes and generates different realizations for the stochastic systems with low cost. Furthermore, we test MGN and GMR-GMUS on the stochastic dataset. In Fig 20 and Table 13, the performance of the proposed model is better than other learning-based models, such that the MGN can not capture the real distribution at all. The result of GMR-GMUS with a sequential model, though looks reasonable in statistics, can only reproduce the same output given the same input, which doesn't reflect the stochasticity.
>
> > **Q2:** It is concerning that the proposed method take all the previous (latent) vectors as input to predict the next state, as opposed to fixed small number of input vectors of MeshGraphNets reported in Section 5 ...
>
> **Response:** MeshGraphNet's in/output are defined on the high-dimensional mesh space instead of low-dimensional vector space. In contrast, the proposed model makes such predictions on latent space with vectors as inputs. Though our model takes more latent vectors as input, the computation is even less than MeshGraphNets during the roll-out stage if the steps are not very large. Since the model can benefit from longer time dependencies, we take all previous steps in the experiment setting. Also, we can use the moving window to reduce the cost further. We did one additional experiment of reducing the number of inputs during the inference time, as shown in supplementary experiments Table 11. The result of reducing the window size from 400 to 150, the degradation in accuracy is not significant. So the model is also applicable to cases with thousands or even longer steps with the sliding window.
>
>
> > **Q3:** Although in Table 1 the reconstruction error of PbGMR-GMUS are compared against GMR-GMUS, it is still unclear how the proposed model would compare against an ablation model whose encoder is GMR-GMUS.
>
> **Response:** Since PbGMR-GMUS improves GMR-GMUS, we prove that PbGMR-GMUS has a better reconstruction error than GMR-GMUS. The following time sequence model can benefit from it. In Table 1, our model performs better on all three datasets compared with [11], whose encoder is GMR-GMUS. To help evaluate the performance against an ablation model with a GMR-GMUS encoder, we listed ablation study results in Table 8 (in new supplementary experiments) and Table 9. In Table 8, residual connection and pivotal positions are necessary for a highly accurate reconstruction of graphs. In Table 9, even with the same attention-based conditional normalizing flow model, the model (Variant 1) with GMR-GMUS as encoder has a worse performance compared with the proposed model.
>
> > **Q4:** The proposed model is compared against baselines which evolve states in original space. There is a possibly missing reference [1]. How does this work compare against it? Is the proposed method also applicable to grid space?
>
> **Response:** Thanks for pointing out this related reference. We added it to the related work. The two works have a similar motivation to make predictions in the latent space for effective computation. However, this reference uses convolutional neural networks to encode the original space, which poses difficulties in applications to irregular mesh spaces. Using CNN to encode images is a very mature technique, but it is difficult to encode graphs. That is also the motivation of this work to propose a general method to encode graphs better. Also, [1] only concentrates on deterministic processes and did not demonstrate on stochastic physics. Still, this reference predicts the PDE process in the latent space, and it will be helpful to readers to know more related works in this area. It is not straightforward to compare the proposed method since we concentrate on systems defined on the irregular mesh. The proposed method is also applicable to the grid space. But the existing CNN already works very well on such tasks.
>
> > **Q5:** In Appendix A.10, Figure 11 is cited, but seemingly wrong.
>
> **Response:** Thanks for pointing it out; we already corrected it in the appendix.  It should be Figure 10.
>
> Following the reviewer's comment improves the quality of the submission significantly. We genuinely appreciate the reviewer's effort and help.

---

> > ### Comment · Reviewer_YdeK · 2023-08-12
> > **Follow-up question**
> >
> > I appreciate that the authors provided detailed explanation for my questions as well as conducted additional experiments. My concerns were satisfactorily addressed, and I was convinced that the proposed method is novel and very effective to solve deterministic and stochastic problems. Although the architecture looks straightforward given the scope of the paper, each of the components is chosen adequately — its adequateness is supported in experiments in the main text and strengthened by additional experiments such as ablation study and comparison against other strong (deterministic) baselines.
> >
> > I still have one follow-up question. What would be the inference time of the proposed model and how does it compare against that of other baselines? Are there any components being bottleneck when performing forward simulation with the proposed model? While the runtime is out of the scope of this paper, it would be beneficial to make sure that the model does not incur significant increase in the computational cost.

---

> > > ### Author Response · Authors · 2023-08-12
> > > **Inference time**
> > >
> > > We really appreciate your careful reading of our response and immediate reply. And we are very glad to know your concerns are addressed. We do think it is important how inference time changes when we unify the deterministic and stochastic problems into the same framework. We compared the inference time of the proposed model with GMUS + Transformer [11] on all four datasets. The results are shown in the following table (unit is seconds, all datasets and models are tested on RTX A6000). There are mainly two steps during inference for the two models: 1. temporal prediction in the latent space. 2. mapping latent vectors into mesh space (decoding).  For the decoding part,  Pb-GMUS has the same inference time as GMUS. And for temporal prediction, we find that the proposed conditional flow takes slightly more time. To include stochastic systems, there is an additional sampling process for the flow model which accounts for additional time. For both models, the decoding part takes more time especially when the original mesh size is large (for example, stochastic flow). So actually the new proposed model doesn't significantly increase the total computational cost. We will add this result to the revised version and we believe such an analysis is beneficial to this work. Thanks for your constructive suggestions!
> > >
> > >
> > >
> > > | Dataset                            | (Pb-)GMUS | Transformer only | Conditional flow (Ours)|
> > > |------------------------------------|-----------|------------------|------------------------|
> > > | Cylinder flow (for 400 time steps) |     2     |             0.93     |       1.66           |
> > > | Sonic flow (for 40 time steps)     |     0.20   |             0.59    |         0.78         |
> > > | Vascular flow (for 250 time steps) |     2.63  |              0.79    |        1.27          |
> > > | Stochastic flow (for 240 time steps) |       6.93    |           0.87       |       1.38           |

---

> > > > ### Comment · Reviewer_YdeK · 2023-08-13
> > > >
> > > > Thank you very much for reporting the inference time. I agree with that the proposed model does not significantly increase the computational cost associated with temporal prediction in latent space. However, it is a bit concerning to me that (Pb-)GMUS takes around 5 folds longer than the latent temporal prediction in the worst case. This is problematic especially when we need to decode the latent vector back every time steps, as you provided in the videos. How do you expect the choice of the model’s architecture or hyperparameters to change, if we want to decrease the decoding time?

---

> > > > > ### Author Response · Authors · 2023-08-13
> > > > > **Parallel computing during decoding**
> > > > >
> > > > > Thanks for your concerns. It should be noticed that in Table 4 (Appendix), the stochastic dataset contains 22500 nodes, much larger than other datasets. We find that the temporal prediction in latent space doesn't increase compared with other datasets. This allows the model to do parallel computing during the decoding. It should be noticed that other roll-out models such as MGN on the original space, can not do such parallel computing and suffer from the increase of the mesh number since they also need to run GNNs at each step. For (Pb-)GMUS, there is no temporal relation among each step. The model is only responsible for mapping all latent vectors back to the original meshes. These computations can be done in parallel. Still on a single GPU (RTX A6000), the memory allows us to divide decoding into 5 independent processes, and the overall decoding time only takes 1.57 seconds. Thanks to the fast inference time for temporal prediction, we can do parallel computing for the heavy computation part. We think this is also an advantage of the proposed model for huge physical datasets. And we appreciate your insightful questions. We will add this discussion to the revised manuscript.

---

> > > > > > ### Comment · Reviewer_YdeK · 2023-08-13
> > > > > > **Strong response from the authors**
> > > > > >
> > > > > > Thank you for the detailed explanation. In light of providing additional experimental results and detailed explanation, I increased my score (Strong accept). The authors’ rebuttal is satisfactory to my concerns and made their claims much more solid. The authors also addressed my additional concerns on computational cost, which are basically out of the scope of the paper, and provided more explanation about how to alleviate the issue and even showed that the inference time is comparable to other strong baselines. I think this is a good paper and above the acceptance threshold. I believe the paper provides value to the learning simulation community.
> > > > > >
> > > > > > I appreciate for the authors’ all efforts during the rebuttal period. I really enjoyed the discussion!

---

> > > > > > > ### Author Response · Authors · 2023-08-13
> > > > > > > **Thanks**
> > > > > > >
> > > > > > > We really appreciate your time and efforts on our submission. We also really enjoy such discussion and believe it is very helpful for improving our manuscript. Thank you so much for the support of our submission!

---

### Official Review · Reviewer_fgSk · 2023-07-06

**Soundness:** 4 excellent
**Presentation:** 4 excellent
**Contribution:** 4 excellent
**Rating:** 8
**Confidence:** 5

**Summary:**

The authors present a new approach for modeling fluid dynamics. The approach involves using GNNs to derive global latent space representations and construct a transformer-based conditional generative models for the dynamics. The resulting model is able to generate stochastic predictions from given initial conditions and system parameters at inference time. The proposed framework is compared against some competitive baselines on a few deterministic and stochastic benchmarks.

**Strengths:**

*  This work seeks to perform probabilistic modeling of high-dimensional dynamical systems originating from parametric PDEs. This is a very important problem as traditional methods are typically quite expensive and lacking ways to quantify uncertainties. The method proposed here is a nice attempt that addresses both challenges.
* The method presented is very flexible and easily adapts to various problems and configurations. Because of the complexity of the cases, it has the potential to serve as a fast surrogate for modeling turbulence useful for inner-loop applications.
* The introduced regeneration learning framework is quite novel as it effectively integrates different blocks (graph representation learning, attention-based model, normalizing flows). Such formulation offers an efficient and attractive alternative to directing learning long-time spatiotemporal fields on the original mesh. Moreover, the framework can handle deterministic/stochastic systems in a unified way.
* The presentation is very clear, logical and easy to understand in general. The plots are of high quality and informative.
* Claims are backed by strong and convincing empirical evidence. The examples in the numerical experiments section demonstrate sufficient complexity and variety to help justify the value of the proposed method.

**Weaknesses:**

* A number of minor typos in texts and figures (see section below).
* More discussions on the limitations would strengthen the paper further.

**Questions:**

Questions
* How much of the improvements can be attributed to using *pivotal positions* and how much to using a different architecture for the latent dynamical model? It would be nice to have some ablation studies
* What is the advantage of using the RealNVP vs other generative models (e.g. there is a number of normalizing flows [here](https://github.com/VincentStimper/normalizing-flows))?

Comments
* Ln 168 - I believe it’s inaccurate to say the geometry of the graph is not included in the latent vector - it’s really embedded implicitly in the representation.
* Ln 243 - “.. potentially improving the interpretability of deep learning systems” - UQ and interpretability are different concepts.

Typos
* Equation (1), $j$ is not defined in text. Based on what follows, it can be a stochastic operator as well?
* Ln 95 - functions that *are* parameterized by neural networks
* Equation (5) - consider not using $p$ for position to avoid confusion with probability densities?
* Typeset errors on ln 219 and 289?
* Figure 3 - axis labels are missing
* Table 3 - what is referred to by “Flow” is your model? I don’t believe this is referenced elsewhere in the text.
* In appendix, Ln 589 - the results are in Figure 10 not 11

**Limitations:**

The authors do present ablation studies showing that a certain number of *pivotal positions* are required. Other potential areas to discuss are scalability, applicability to more complex systems, sampling requirements and parameter extrapolations.

---

> ### Author Rebuttal · Authors · 2023-08-10
>
> Thanks for the reviewer's supportive evaluation of our work. We are trying our best to address your concerns with the following answers.
>
> > **Q1:**: A number of minor typos in texts and figures (see section below).
>
> **Response:** Thanks for your careful reading. We already fixed them in the revised version.
>
> > **Q2:**:More discussions on the limitations would strengthen the paper further.
>
> **Response:** Thanks for this concern. We agree it is necessary to discuss the limitations of the proposed model. Please refer to global rebuttal **Q3** for details.
>
> > **Q3:** How much of the improvements can be attributed to using pivotal positions and how much to using a different architecture for the latent dynamical model? It would be nice to have some ablation studies
>
> **Response:** Thanks for asking this question. We already made more comprehensive ablation studies and the reviewer can refer to our global rebuttal and new supplementary experiment table 8 and table 9. Ablation studies show that pivotal positions are helpful to improve reconstruction accuracy. And the proposed conditional flow model performs better against a single transformer model with the same encoder-decoder structure. We believe these new ablation studies are helpful in identifying the use of each component in our new design.
>
>
>
>
> > **Q4:** What is the advantage of using the RealNVP vs other generative models (e.g. there is a number of normalizing flows here)?
>
> **Response:** Thanks for the reviewers' insightful question on different normalizing flows models and providing the links. The key design for the time series prediction model is to adopt a transformer-based encoding-decoding structure to capture temporal conditions as well as global physical parameters. The reason we adopt the flow model is that there are fewer parameters to compute and it is easier to get samples at each step compared with other probabilistic models such as Gaussian distribution. We think other normalizing flows are also feasible here. In supplementary experiment table 12, we test another normalizing flow model MAF, and find the result is comparable to RealNVP. It indicates that the proposed framework is flexible.
>
>
> > **Q5:** Ln 168 - I believe it’s inaccurate to say the geometry of the graph is not included in the latent vector - it’s really embedded implicitly in the representation.
> Ln 243 - “.. potentially improving the interpretability of deep learning systems” - UQ and interpretability are different concepts.
>
> **Response:** Thanks for reading our script carefully and giving constructive feedback. We agree that the geometry is implicitly embedded into the representation. Moreover, we agree that UQ and interpretability are different concepts. The current model can capture the UQ for deterministic systems and generate physical realizations for stochastic systems. We already fixed these in the revisions.
>
> We've made updates based on the feedback provided, and we believe that these changes substantially improve the manuscript. We're grateful for the valuable comments and thank you for your time and expertise.

---

> > ### Comment · Reviewer_fgSk · 2023-08-17
> >
> > Thank you for your detailed response. The additional ablation studies are very informative - it is nice to see the effects of pivotal positions, residual connections, and conditional normalizing flows in isolation. And the additional baselines further justify the effectiveness of the proposed model.
> >
> > I have another clarification question: regarding errors (both RMSE and CRPS) reported, are they based on the mean of several realizations of your model prediction? If so, what is the ensemble size?

---

> > > ### Author Response · Authors · 2023-08-18
> > >
> > > Thank you so much for carefully reading our rebuttal and giving us feedback! We are glad our rebuttal helped the reviewer better understand our novelty and significance of unifying predictions of deterministic and stochastic dynamics in one model. Moreover, our proposed framework can achieve this without introducing too much computational overhead during inference. As for rmse, we reported the mean rmse error with ensemble size 10. For CRPS, we generated 30 different realizations for the stochastic dataset. We appreciate the reviewer's clarification question and will add this to the manuscript.

---

> > > > ### Comment · Reviewer_fgSk · 2023-08-19
> > > >
> > > > Thanks for the clarifications. I just want to kindly note that all these details are important for reproducing purposes so should be included in the revised version.
> > > >
> > > > I read all the reviews and find the concerns addressed very well. The efforts made during the rebuttal period substantiate the authors' claims. I believe the proposed framework is a very promising approach in modeling complex dynamics. Overall, I am confident in keeping the strong accept (SA) rating.

---

### Author Rebuttal · Authors · 2023-08-09

Dear reviewers,

We would like to express our gratitude for your constructive feedback on our manuscript. The insights provided have been invaluable in refining our work. We have uploaded one-page supplementary experimental results in response to your comments.

Here we want to highlight several contributions mentioned in the reviews:

- The proposed method is simple and applicable to wide range of PDE-simulation problems with discretized domain. Wide variety of experiments are conducted, and both quantitative and qualitative evaluations are provided.

- This paper introduces the first learning-based probabilistic model that simulates fluid systems upon the mesh topology.

- This work seeks to perform probabilistic modeling of high-dimensional dynamical systems originating from parametric PDEs. This is a very important problem as traditional methods are typically quite expensive and lacking ways to quantify uncertainties. The method proposed here is a nice attempt that addresses both challenges.

- The introduced regeneration learning framework is quite novel as it effectively integrates different blocks (graph representation learning, attention-based model, normalizing flows). Such formulation offers an efficient and attractive alternative to directing learning long-time spatiotemporal fields on the original mesh. Moreover, the framework can handle deterministic/stochastic systems in a unified way.

- The simulation of dynamics is achieved by a probabilistic method using generative models, which rollout vivid predictions for long-term predictions. The proposed method is able to handle both deterministic and stochastic fluid dynamics thanks to the probabilistic model and achieves superior performance over previous methods.

Also, based on constructive suggestions, we conducted several ablation studies to identify the use of each component and compare the results on stochastic physics with the learning-based model MeshGraphNet[3] and GMR-GMUS + transformer[11]. Here, we want to solve the issues the reviewers are concerned mostly.

> **Q1** There should be ablation studies to identify the use of each component. (Reviewer YdeK, 9WF1, and 8v2N)

**Response:**  We appreciate the reviewers' suggestions and believe such ablation studies are helpful. We first check the velocity reconstruction error on the Backward-facing step flow dataset with or without residual connection and pivotal positions. The result of Table 9 indicates each component can improve the graph reconstruction accuracy. We also test the influence of PbGMR-GMUS and the attention conditional flow model to the final performance of the deterministic task. Table 10 in supplemental experiments indicates both the encoder-decoder part and the flow model is necessary for the success of the whole framework.

> **Q2** The model should be compared against other learning-based models, such as MeshGraphNet and GMR-GMUS, on the stochastic dataset to prove the superiority of the proposed approach. What are the true challenges of the prediction task for stochastic fluid systems?(Reviewer 9WF1, YdeK)

**Response:** It is very hard for the previous models to predict stochastic mesh-based physical systems. Defining a probabilistic model on such sequential data is not easy. If we use a Gaussian distribution for the targeted system, which contains $N$ meshes, at each time step, the model should predict $N$ means and $N$x$N$ covariance. Also, the model should take temporal- and spatial- dependencies into consideration. Both make the computation too heavy to be applicable. Also, sampling from such distribution is computation-consuming. Most of the previous models, such as MeshGraphNet[3], and GMR-GMUS decoding-only[11], are not probabilistic models, so they can not be applied to any stochastic system. Given the same initial state and physical parameters, they always produce the same prediction. In contrast, as a probabilistic model, the proposed PbGMR-GMUS + conditional flow model is trying to fit such a stochastic process and get various samples with the same physical statistics. To demonstrate these, we test MGN and GMR-GMUS on the stochastic dataset. In Table 13, we can see that the performance of the proposed model is better than other learning-based models on several metrics. In Figure 20, we visualize the mean and variance of velocity from each model, and find the MGN can not capture the real distribution at all. The result of GMR-GMUS model, though looks reasonable in statistics, can only produce the same output given the same input condition, which doesn't reflect the stochasticity of the last dataset.  In Figure 21, we get two samples from GMR-GMUS[11], and find the two samples are exactly the same.

> **Q3** More discussions on the limitations would strengthen the paper further. (Reviewer 9WF1, fgSk)

**Response:** We discussed the limitation of the proposed model in Appendix, section A.12. Reviewer YdeK
also mentioned that the limitations are discussed in the appendix. The model can not capture an accurate distribution for stochastic systems close to boundary areas and regions where flow is not fully developed. (Details and figures are also in this part) We agree that readers may ignore such discussions, and we will add some analysis to the main paper and give a reference to the appendix.

In conclusion, we have incorporated the insightful feedback provided by the reviewers, enhancing the overall quality of our paper. We sincerely value the time and effort invested in reviewing our work and believe that the revised manuscript and supplementary experiments address the concerns raised.

---

> ### Author Response · Authors · 2023-08-21
> **Thank you for your commitment**
>
> Since it approaches the end of the author-reviewer discussion period, we would like to sincerely thank the reviewers for their invaluable contributions during the review and rebuttal period.
>
> Your insightful questions and engaging discussions have been instrumental in refining and strengthening the quality of our work. Your thoughtful inquiries helped us clarify our research and pushed us to delve deeper into our study. We are grateful for your feedback and will incorporate your suggestions in our manuscript. Thank you once again for your time and efforts in reviewing our submission.

---

### Comment · Area_Chair_ap82 · 2023-08-12
**Video demo from authors**

The authors have kindly shared video demonstration of their predictions at https://drive.google.com/drive/folders/1bk8y0DOLrHHv42QZKEBduL-fMp9__-uH?usp=sharing

---

### Decision · Program_Chairs · 2023-09-21

**Decision:**

Accept (spotlight)

**Comment:**

Reviewers agreed about paper's excellent presentation of integrating many different ideas to achieve compelling results on the hard problem of learning stochastic dynamics from LES data. I especially liked the inclusion of alternative evaluation metrics other than normalized RMSE in this domain and will go a long way in helping improve comparisons between different methods in this domain.